# Dynamic movement of the Golgi unit and its glycosylation enzyme zones

Akihiro Harada [1,13] ✉, Masataka Kunii[1,13], Kazuo Kurokawa [2,13,14], Takuya Sumi [1,13], Satoshi Kanda[1], Yu Zhang[1], Satomi Nadanaka [3], Koichiro M. Hirosawa [4], Kazuaki Tokunaga[5], Takuro Tojima [2], Manabu Taniguchi[1], Kenta Moriwaki [1], Shin-ichiro Yoshimura[1], Miki Yamamoto-Hino[6], Satoshi Goto [6], Toyomasa Katagiri[7], Satoshi Kume [8], Mitsuko Hayashi-Nishino[9], Miyako Nakano[10], Eiji Miyoshi [11], Kenichi G. N. Suzuki [4,12], Hiroshi Kitagawa [3] & Akihiko Nakano [2]

Knowledge on the distribution and dynamics of glycosylation enzymes in the Golgi is essential for better understanding this modification. Here, using a combination of CRISPR/Cas9 knockin technology and super-resolution microscopy, we show that the Golgi complex is assembled by a number of small 'Golgi units' that have 1-3 µm in diameter. Each Golgi unit contains small domains of glycosylation enzymes which we call 'zones'. The zones of N- and O-glycosylation enzymes are colocalised. However, they are less colocalised with the zones of a glycosaminoglycan synthesizing enzyme. Golgi units change shapes dynamically and the zones of glycosylation enzymes rapidly move near the rim of the unit. Photobleaching analysis indicates that a glycosaminoglycan synthesizing enzyme moves between units. Depletion of giantin dissociates units and prevents the movement of glycosaminoglycan synthesizing enzymes, which leads to insufficient glycosaminoglycan synthesis. Thus, we show the structure-function relationship of the Golgi and its implications in human pathogenesis.

In the Golgi complex, various types of sugar chains, such as N-type, O-type or mucin type, and sulphated glycosaminoglycan (GAG) are covalently linked to proteins, and these sugar chains are important in determining the structure and function of glycosylated proteins. Different proteins acquire unique glycosylation. However, many topics concerning glycosylation remain unclear, such as how different types of glycosylation reactions are rapidly and simultaneously performed and how the number of sugar chains is determined. To date, the distribution of glycosylation enzymes in the Golgi apparatus has been well examined for *cis-trans* distribution[1–4], but less is known about where they localise within a single cisterna. Recently, dissociated Golgi fragments were observed by Airyscan confocal microscopy combined

[1]Department of Cell Biology, Graduate School of Medicine, Osaka University, Suita, Osaka, Japan. [2]Live Cell Super-Resolution Imaging Research Team, RIKEN Center for Advanced Photonics, Wako, Saitama, Japan. [3]Laboratory of Biochemistry, Kobe Pharmaceutical University, Kobe, Hyogo, Japan. [4]Laboratory of Cell Biophysics, Institute for Glyco-core Research (iGCORE), Gifu University, Gifu, Gifu, Japan. [5]NIKON SOLUTIONS CO., LTD., Tokyo, Japan. [6]Department of Life Science, Rikkyo University, Toshima-ku, Tokyo, Japan. [7]Laboratory of Biofunctional Molecular Medicine, National Institute of Biomedical Innovation, National Institutes of Biomedical Innovation, Health and Nutrition, Ibaraki, Osaka, Japan. [8]Laboratory for Pathophysiological and Health Science, RIKEN Center for Biosystems Dynamics Research, Kobe, Japan. [9]SANKEN (The Institute of Scientific and Industrial Research), Osaka University, Ibaraki, Osaka, Japan. [10]Graduate School of Integrated Sciences for Life, Hiroshima University, Higashi-Hiroshima, Hiroshima, Japan. [11]Department of Molecular Biochemistry and Clinical Investigation, Osaka University Graduate School of Medicine, Suita, Osaka, Japan. [12]Division of Advanced Bioimaging, National Cancer Center Research Institute, Tokyo, Japan. [13]These authors contributed equally: Akihiro Harada, Masataka Kunii, Kazuo Kurokawa, Takuya Sumi. [14]Deceased: Kazuo Kurokawa. ✉e-mail: aharada@acb.med.osaka-u.ac.jp

with image averaging, revealing that a number of Golgi proteins are distributed relative to the centre of the fragmented Golgi[3,4]. However, the averaging method is not suitable to precisely localise and measure the area occupied by each protein. Additionally, in many cases, the localisation of glycosylation enzymes has been investigated by overexpressing the cDNAs of the enzymes of interest because of the lack of proper antibodies. However, an overexpressed protein may occupy a larger area than its endogenous area and affect the dynamics of expressed proteins. For these reasons, the precise endogenous localisation and dynamics of glycosylation enzymes within one Golgi cisternae have not been examined thus far. With the advent of CRISPR/Cas9 technology, we can knock in exogenous tags into the desired genes of any species of organisms to monitor their endogenous localisation and dynamics[5]. At the same time, the development of high-speed super-resolution microscopy SCLIM (super-resolution confocal live imaging microscopy) has made it possible to observe not only the localisation of proteins and lipids at the nanoscale level but also their dynamics by super-resolution live imaging[6–10] in three dimensions. In this work, we demonstrate the basic structure and dynamics of the Golgi and its glycosylation enzymes using various super-resolution microscopies and immunoelectron microscopy.

## Results

### Knockin reveals endogenous localisation of enzymes

Although glycosylation enzymes are essential for the glycosylation of proteins and lipids, their endogenous localisation within the plane of cisternae is poorly understood. To elucidate their endogenous localisation, we knocked in various tags (9x Myc, 3x PA, 3x V5, Halo, and SNAP tags)[11–14] at the C-terminus of glycosylation enzymes (Fig. 1a–c and Supplementary Fig. 1b). PA tag is a 12 amino acid epitope from human podoplanin and is known to have a higher affinity than flag, HA, and myc tags[14]. Insertion at the C-terminus has been frequently used for cDNA transfection because it does not affect the localisation of the glycosylation enzymes in the Golgi[15,16]. We mainly used three enzymes at the early stage of glycosylation in the Golgi, which are reported to localise in the *cis* to *medial* Golgi cisternae (Fig. 1b and Supplementary Figs. 1a and 4a–c). These are an N-glycosylation enzyme, MGAT1 (Alpha-1,3-Mannosyl-Glycoprotein 2-Beta-N-Acetylglucosaminyltransferase)[1,2], an O-glycosylation enzyme, GALNT6 (Polypeptide N-Acetylgalactosaminyltransferase 6)[1,17], and a GAG synthesising enzyme, XYLT2 (Xylosyltransferase 2)[1,18] (Fig. 1b). In addition, we used two enzymes that are necessary at later stages of glycosylation (Fig. 1b and Supplementary Figs. 1a and 4d). One is β4GalT1 (beta4-galactosyl transferase 1), an enzyme synthesising polylactosamine on both N- and O-glycans[1,19,20]. The other is NDST1 (N-Deacetylase And N-Sulfotransferase 1), which catalyses the transfer of sulphate to the nitrogen of glucosamine in a GAG molecule, heparan sulphate[1,21].

We checked whether the PA tag was properly inserted into the locus of the *β4GalT1* and *NDST1* genes because the tag is known to exhibit higher affinity than that of flag or HA tags but is less frequently used previously[14]. The tagged enzymes were identified at the proper molecular weight by genomic PCR and western blotting (Supplementary Fig. 1b–d). We also confirmed that the insertion of the 3xPA and 9xMyc tags did not affect the glycosylation activity of XYLT2 and NDST1, respectively (Supplementary Fig. 2). Using the super-resolution confocal live imaging microscopy (SCLIM)[7,9], we confirmed that the 3xPA tag and the Halo tag did not affect the localisation of GALNT6 (Supplementary Fig. 3a, b). We also tested the validity of knockin by comparing the signals obtained by knockin and cDNA transfection. The signal obtained by knockin was concentrated and punctate within the Golgi, whereas the signal obtained by transfection localised more diffusely and filled throughout the Golgi (Supplementary Fig. 3c). To assess the localisations of the glycosylation enzymes in the Golgi clearly, we dispersed the Golgi by nocodazole, which dissociates the

large Golgi apparatus into smaller Golgi discs, or 'mini-stacks' surrounded by giantin or other golgins (golgin-84 and TMF1)[3] (Supplementary Figs. 3d and 8a). We called these Golgi discs as the Golgi 'units' because they have relatively uniform diameters and they have all sets of glycosylation enzymes examined as described later (Figs. 2d and 4b).

### Comparison of glycosylation enzymes

Given that the localisation of endogenous enzymes is not affected by tags, we compared the localisation of GALNT6, MGAT1, and XYLT2 in the same triple KI cell line (GALNT6-9xMyc + MGAT1-3xPA + XYLT2-3xV5) (Fig. 1d, e). All three enzymes occupied discrete small areas within a Golgi unit (Fig. 1e). The correlation coefficient of GALNT6 and MGAT1 was significantly higher than that of GALNT6 and XYLT2 in nocodazole-treated cells (Fig. 1f). This observation was confirmed in nocodazole-untreated cells (Fig. 1f). A high correlation coefficient (0.7-0.8) between GALNT6 and MGAT1 indicates that they are almost colocalised because the correlation coefficients between the different tags within the same molecule and the one between different dyes (J549 and J646) which bind to Halo tag are around 0.8 (Fig. 2g and Supplementary Fig. 3b). However, GALNT6 and XYLT2 overlap to a lesser extent (correlation coefficient: 0.5–0.6) (Fig. 1f).

To determine their localisations in one unit of the Golgi, we compared their localisations in side views and en face views[3] of the same mini-stack (Fig. 1g). When GALNT6, MGAT1, and XYLT2 were compared with giantin, these four proteins localised almost at the same level in side views, which indicates that they localised almost at the same cisternae (Fig. 1g). For en face views, although GALNT6 and MGAT1 colocalised well, GALNT6 and XYLT2 did not (Fig. 1g). As all these glycosylation enzymes occupy small areas, we called these small areas as 'zones' of glycosylation enzymes, as previously designated[8].

### O-glycosylation zones and GAG zones are distinct

The previous result showed that glycosylation enzymes are mainly localised at the centre of the small Golgi units using Airyscan confocal microscopy[3]. Thus, to investigate their localisation at higher resolution, we utilised localisation microscopy, STORM[22,23]. We used giantin for the first dye as a reference for the rim of the Golgi to align the Golgi unit (Supplementary Fig. 3d). Afterwards, we stained one of the glycosylation enzymes (GALNT6, MGAT1, and XYLT2) for the second dye to measure the correlation coefficient between the enzyme and giantin; this was performed to determine their localisation relative to the rim of the Golgi unit. In nocodazole-treated cells, GALNT6 and MGAT1 appeared to localise diffusely and colocalise with giantin well. In contrast, XYLT2 localised as small puncta, some of which colocalised with giantin, while others were distributed away from giantin (Fig. 2a). In support of this observation, the correlation coefficient between GALNT6/MGAT1 and giantin was higher than that between XYLT2 and giantin (Fig. 2b). This is consistent with the results obtained by SCLIM, i.e., the correlation coefficient between the three dimensional volume occupied by GALNT6 and MGAT1 is higher than that between GALNT6 and XYLT2. We obtained a similar result in isolated units in nocodazole untreated cells (Fig. 2b). Thus, we mainly used the results from nocodazole untreated cells because it more likely reflected the physiological cellular condition.

Additionally, we examined whether all three glycosylation enzymes were localised in one unit by STORM. All the enzymes were detected in the same unit, although the amount varied from one unit to another (Fig. 2d). Based on this finding, one Golgi unit appears be a minimal functional unit. This is in contrast with the dissociated Golgi in *Drosophila*, which frequently exhibits only one type of glycosylation enzyme in one small Golgi[24]. Since some XYLT2 zones appeared as smaller puncta compared with those of GALNT6 and MGAT1 (Fig. 2a), we calculated the areas based on the images of STORM by image processing[25,26] (Fig. 2c). As the areas of GALNT6 zones and XYLT2 zones were significantly

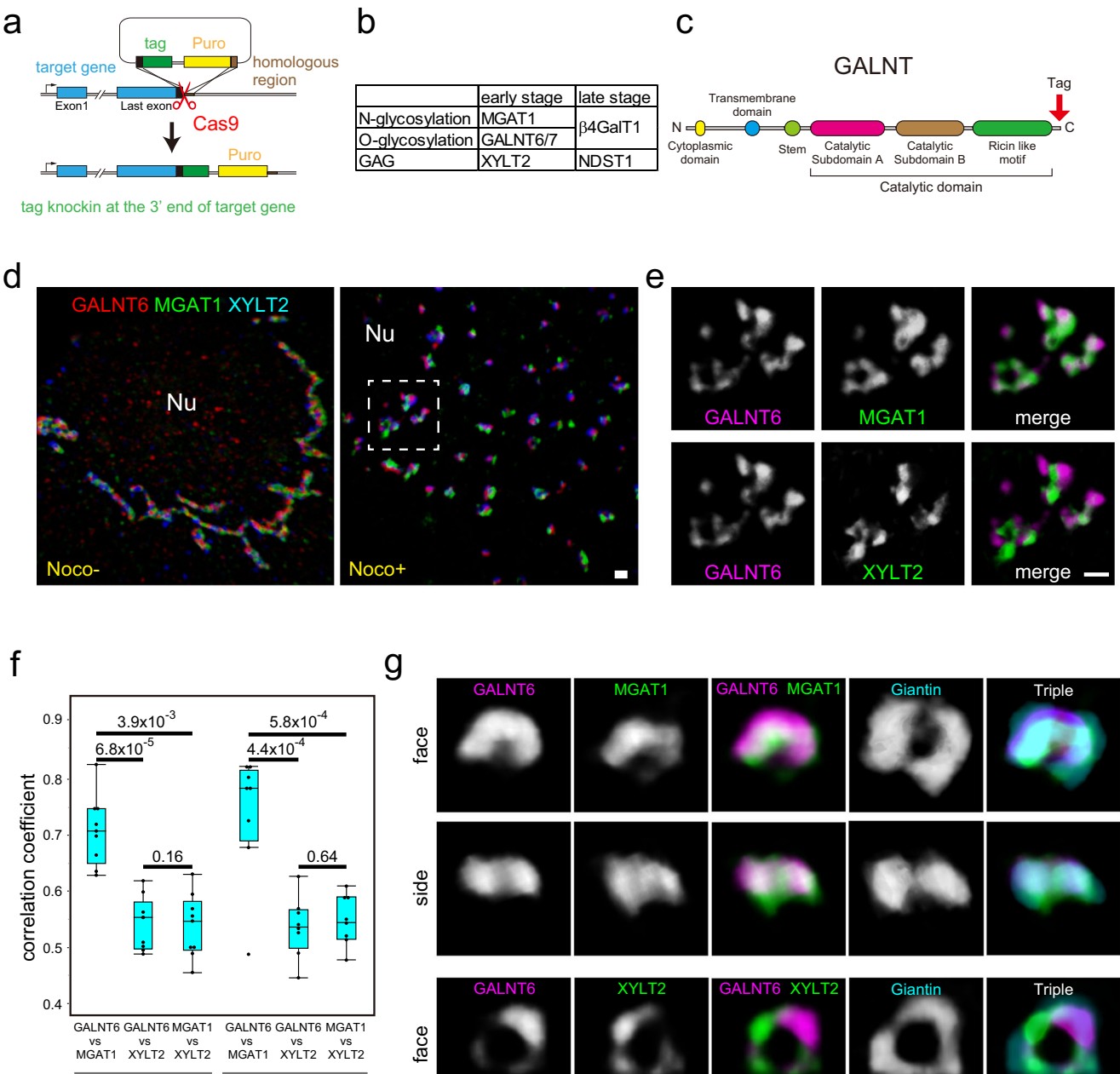

**Fig. 1 | Distribution of glycosylation enzymes. a** Schematic representation of knockin of tags into the exon of glycosylation enzymes. **b** Glycosylation enzymes used for knockin in this study. **c** Schematic representation of the GALNT enzyme. A tag was inserted after the catalytic domain at the C-terminus. **d**, **e** Multicolour SCLIM 3D imaging of KI cells stained with antibodies against various tags. **d** Distribution of GALNT6, MGAT1, and XYLT2 in a nocodazole-treated and nocodazole-untreated triple KI cell line (GALNT6-9xMyc + MGAT1-3xPA + XYLT2-3xV5) at low magnification. Left: a nocodazole untreated cell, Right: a nocodazole treated cell. **e** Magnified views of the dashed rectangle in the right panel of **d**. Upper panels: A comparison between GALNT6 and MGAT1. Lower panels: A comparison between GALNT6 and XYLT2. Nu: nucleus. Bars, 1 μm. **f** 3D colocalisation analysis between two enzymes in triple KI cells with or without nocodazole. Statistical

significance was determined by two-tailed unpaired t test. Number of cells: 9 (nocodazole-) and 8 (nocodazole + ). *P* values are depicted in the graph. Boxes represent 25% and 75% quartiles, lines within the box represent the median, and whiskers represent the minimum and maximum values within 1.5x the interquartile range. **g** Three-colour SCLIM 3D imaging of isolated Golgi units in KI cells stained with antibodies against Myc tag, PA tag, and giantin. The upper two lines of panels show the Golgi in GALNT6-9x Myc+MGAT1-3xPA double KI cells. The lower two lines of panels show the Golgi in GALNT6-9x Myc+XYLT2-3xPA double KI cells. face: en face view, side: side view. The same Golgi unit is observed from different angles in the top two rows and the bottom two rows. Bar, 1 μm. Images are representative of at least three independent biological replicates. Imaging modalities and acquisition parameters of SCLIM are described in the materials and methods.

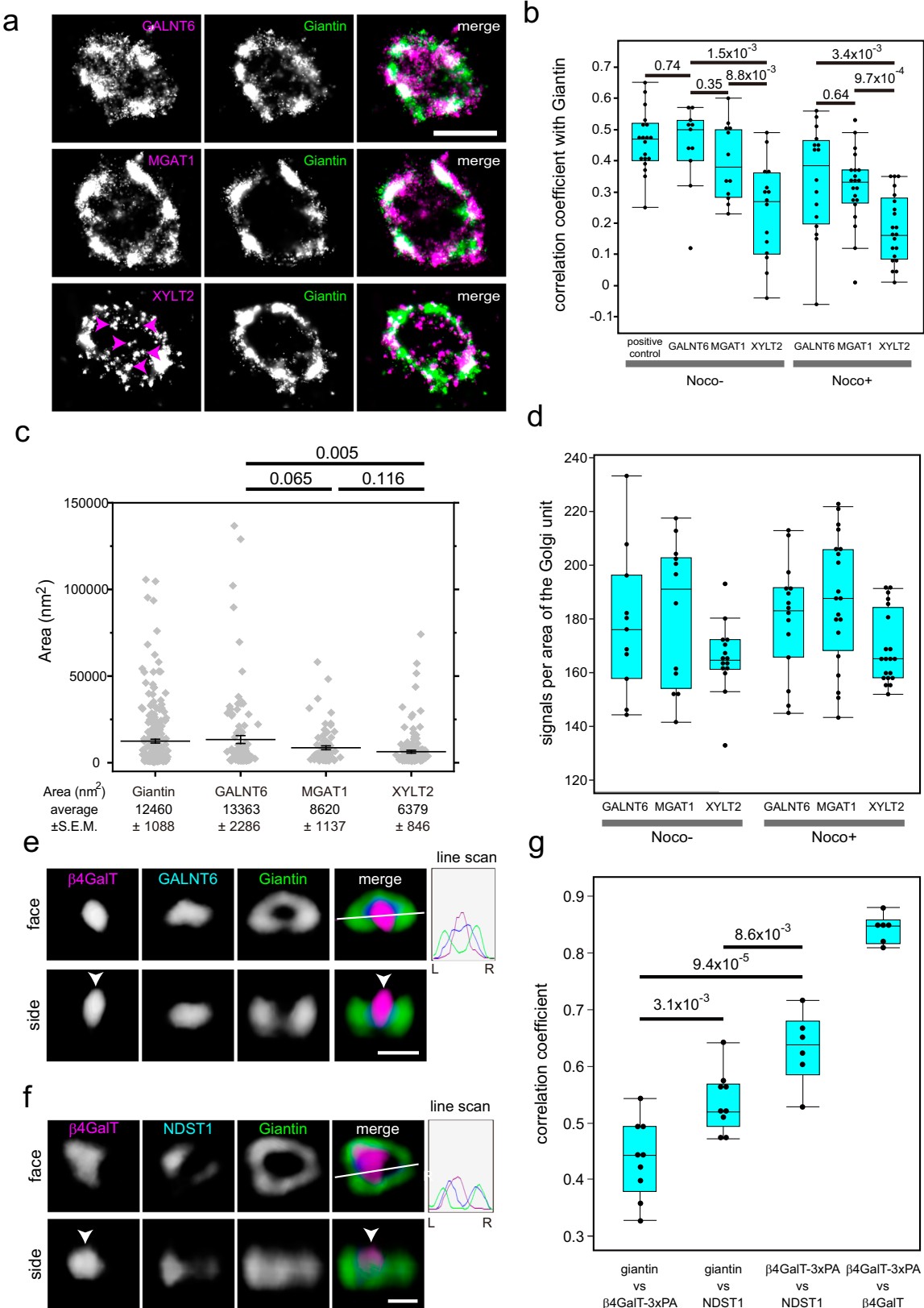

different, the zones of the O-glycosylation enzyme and the GAG synthesising enzyme appeared to exhibit different characteristics.

To exclude the possibility that the reduced correlation coefficient between GALNT6 and XYLT2 came from their localisation in distinct cisternae, we examined their localisation further by immunoelectron microscopy. First, we checked the localisation of GALNT6. By

examining GALNT6 with a *cis*-Golgi marker GM130 through immunoelectron microscopy, we determined that GALNT6 signals were localised predominantly in the darkly stained GM130-positive *cis* cisternae and the CGN; however, they were also identified in the *medial* and *trans* cisternae to a lesser extent (Supplementary Fig. 4a, b), indicating that GALNT is relatively broadly distributed from the CGN to

**Fig. 2 | Distribution of glycosylation enzymes observed by STORM and SCLIM.** **a** STORM 2D imaging of the isolated Golgi unit in en face view of nocodazole-treated KI cells stained with antibodies against the PA tag and giantin. Upper panels: GALNT6-3xPA KI cells. Middle panels: MGAT1-3xPA KI cells. Lower panels: XYLT2-3xPA KI cells. Arrowheads indicate punctate staining of XYLT2. Bar, 1 μm. **b** STORM 2D colocalisation analysis among KI cells treated with and without nocodazole. Statistical significance was determined by two-tailed unpaired t test. In **b**–**d**, number of cells: 4 (GALNT6, MGAT1: nocodazole-), 5 (XYLT2: nocodazole-) and 6 (nocodazole + ). In **b**, **c,**, **g**, *P* values are depicted in the graph. In **b**, **d**, **g**, boxes represent 25% and 75% quartiles, lines within the box represent the median, and whiskers represent the minimum and maximum values within 1.5 x the interquartile range. **c** Distribution of areas of the zones of various glycosylation enzymes and giantin. The areas of individual zones (grey dots) in STORM images were determined using the Voronoi-based segmentation method (see Methods section for details). Error bars represent the standard error of the mean (SEM). Statistical significance was determined by Welch's two-tailed t test. N = 263, 107, 81, and 148 for giantin, GALNT6, MGAT1, and XYLT2, respectively. **d** Distributions of signal intensities in one Golgi unit of cells treated with and without nocodazole. **e**, **f** SCLIM 3D imaging of GALNT6-3xPA KI cells stained with antibodies against β4GalT1, PA tag, and giantin (**e**) and that of β4GalT1-3PA + NDST1-9xMyc KI cells stained with antibodies against the PA tag, Myc tag, and giantin (**f**). Arrowheads denote the localisation of β4GalT at the centre of the Golgi unit. The fluorescence intensity profile from left (L) to right (R) in the right upper panels is indicated on the right. Bars, 1 μm. **g** SCLIM 3D colocalisation analysis among β4GalT1, NDST1, and giantin. The right bar shows the colocalization coefficient of PA tags with β4GalT1 in β4GalT1-3xPA KI cells, which only have KI alleles (Supplementary Fig. 1c, d). Each point represents the value from one cell. Thus, the number of cells is more than 6. Statistical significance was determined by two-tailed unpaired t test. Imaging modalities and acquisition parameters of SCLIM are described in the materials and methods.

the *medial* Golgi. Then, by examining GALNT6 and XYLT2 through immunoelectron microscopy, XYLT2 signals were identified only in the electron-dense GALNT6-positive cisternae rather than GALNT6-negative cisternae (Supplementary Fig. 4c), which confirmed the result obtained by SCLIM (Fig. 1g).

### Comparison of early/late-stage glycosylation enzymes

We next compared the localisations of the early- and late-stage glycosylation enzymes (Fig. 2e) by SCLIM. As β4GalT1 is a late-stage enzyme for both N- and O-glycosylation (Fig. 1b), it is not necessary to use STORM to discriminate zones for N- and O-glycosylation zones further. β4GalT1 is localised predominantly at the centre of the Golgi unit (Fig. 2e, f, g); in contrast, GALNT6, an early-stage enzyme for O-glycosylation (Fig. 1b), localised at or near its rim (Fig. 2e). Next, to know the relative localisations of two different late-stage enzymes on the same cisternae (Supplementary Fig. 4d), we compared the localisations of β4GalT1 and NDST1, a late-stage enzyme for GAG synthesis (Fig. 1b). β4GalT1 was shown to localise closer to the centre of the unit than NDST1 by comparing their correlation coefficient against the rim marker, giantin (Fig. 2f, g).

From the previous result, XYLT2, an early-stage enzyme for GAG synthesis, localises near the rim of the Golgi units (Fig. 1g). As NDST1 also localises closer to the rim compared to β4GalT1, we examined whether XYLT2 and NDST1 colocalise in en face view. However, they did not colocalize, namely, the zone of NDST1 was not straight above that of XYLT2 (Supplementary Fig. 4e), indicating their independent localisation within a unit. This seems to exclude the possibility that one Golgi unit is further divided into one smaller unit specialised for N/O-glycosylation and the other one specialized for GAG synthesis connected side by side.

To determine whether the substrates, glycosylated proteins, can go through the glycosylation zones, we traced the transport of proteoglycan by the RUSH system[27]. We constructed a RUSH construct of syndecan2 (SDC2), a substrate protein for GAG modification, and introduced it into the XYLT2 and NDST1 double KI cell line. As expected, the correlation coefficient between SDC2 and the early-stage enzyme XYLT2 peaked earlier than that between SDC2 and the late-stage enzyme NDST1 (Supplementary Fig. 4f, g). This indicates that the glycosylation substrates enter the glycosylation zones according to the order of the glycosylation reaction.

### Comparison of early-stage glycosylation enzymes with COPI vesicles

As Golgi resident glycosylation enzymes are known to be recycling in COPI vesicles, some of the population of enzymes will reside in these vesicles. To test whether these enzymes are localised in COPI vesicles, we first compared the localisation of GALNT6 and XYLT2 with that of βCOP (Fig. 3). A part of both GALNT6 and XYLT2 are localised with βCOP outside of a giantin ring (Fig. 3a, b). The extent of colocalization between GALNT6 vs βCOP is similar to that between XYLT2 vs βCOP (Fig. 3c). To know the localisation of these enzymes with vesicles near the Golgi cisternae, we performed immunoelectron microscopy and observed their localisation in en face view of the Golgi. We observed XYLT2 is localised on the vesicle near the cisternae (Fig. 3d). These results indicates that some of the enzymes are localized in the COPI vesicles.

### Golgi ribbon is assembled by dynamic units

Usually, the Golgi complex is localised as clusters or ribbons near the nucleus in interphase in the absence of nocodazole. When we examined the interphase Golgi in three dimensions by SCLIM, we observed that the Golgi was an assembly of units marked by giantin with diameters of 1-3 μm (Fig. 4a). Although glycosylation enzymes (GALNT6 and XYLT2) were mainly localised near the rim of the units, they were sometimes localised inside of them, and both were contained in every unit. The diameter of these units was almost the same whether they were clustered or not (Fig. 4b). These observations indicate that the units in the cluster are the Golgi units as well. They are usually connected side by side with each other or connected by tubules (Fig. 4a, Supplementary Movie 1). We identified that the Golgi was composed of units in other human cell lines, such as HeLa and T47D, and murine primary cells, such as mouse embryonic fibroblasts and neurons (Fig. 4c). Therefore, the Golgi is composed of Golgi units regardless of their species and tissues. Sometimes we observed elongated units in some type of cells. Considering the dynamic nature of the units, this may due to the elongation of the units caused by tension between them and/or reflect the transient shape after deformation and fusion as shown in Fig. 5 and supplementary Movies 4–8. To know the effect of nocodazole, we measured the sizes of the Golgi units of Caco2 cells and HeLa cells with or without nocodazole, and the sizes of the units decreased in the presence of nocodazole (Supplementary Fig. 5a, b).

When we treated Caco2 cells with nocodazole and observed by live imaging by SCLIM, the Golgi ribbon was broken down into large structures which were likely to correspond to the Golgi units and small fragments (Supplementary Movie 2). By blocking the function of microtubule-dependent motor protein, cytoplasmic dynein, the Golgi became fragmented, suggesting the critical role of microtubules in the assembly of the Golgi[28,29]. Thus, Golgi units and the tubules between them are physically fragmented by acute and harsh microtubule-depolymerising effect of nocodazole. Therefore, to know assembly of the Golgi ribbon from the Golgi units clearly, we observed the recovery process of the Golgi ribbon after nocodazole washout because it is slower and milder than nocodazole addition. We were able to observe the re-establishment of the Golgi ribbon by connection of multiple units through tubules between them (Supplementary Fig. 5c, Supplementary Movie 3).

When we transfected *mNeonGreen-giantin* and observed the dynamics of the Golgi units by Airyscan confocal microscopy, we

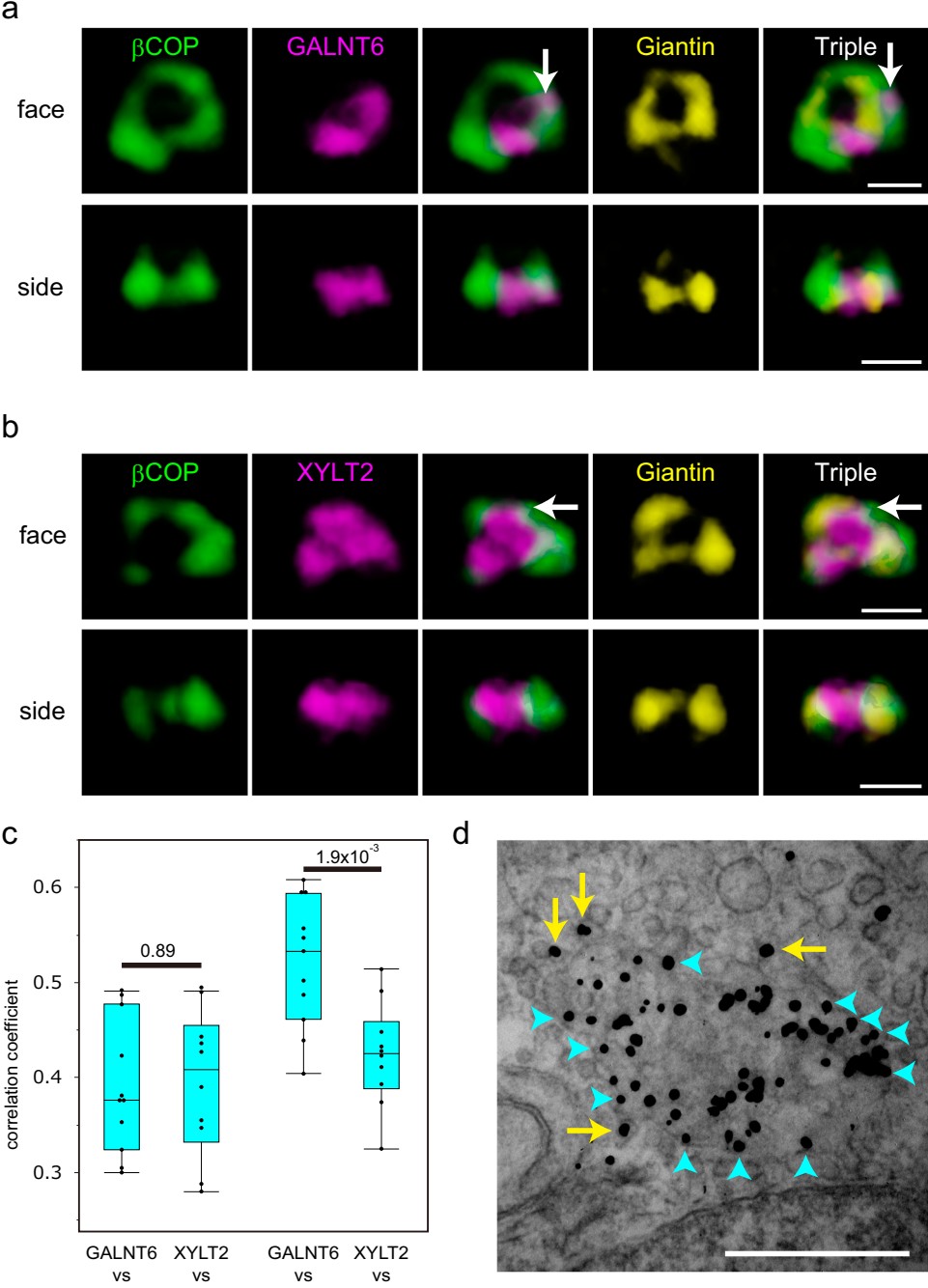

**Fig. 3 | Distribution of glycosylation enzymes with COPI. a, b** Three-colour SCLIM 3D imaging of isolated Golgi units in KI cells stained with antibodies against βCOP, giantin, and PA tag. The upper two lines of panels show the Golgi in a GALNT6-3xPA KI cell. The lower two lines of panels show the Golgi in a XYLT2-3xPA KI cell. face: en face view, side: side view. The same Golgi unit is observed from different angles in Fig. 3a and 3b. βCOP signals colocalised with GALNT (Fig. 3a) and XYLT2 (Fig. 3b) outside of giantin are shown by arrows. Bar, 1 μm. Images are representative of at least three independent biological replicates. Imaging modalities and acquisition parameters of SCLIM are described in the materials and methods. **c** 3D colocalisation analysis between GALNT6/XYLT2 and βCOP (left two lanes) and GALNT6/XYLT2 and giantin (right two lanes) in cells without nocodazole. The number of cells: 10 (left two lanes) and 11 (right two lanes). Statistical significance was determined by two-tailed unpaired t test. Each point represents the value from one cell. *P* values are depicted in the graph. Boxes represent 25% and 75% quartiles, lines within the box represent the median, and whiskers represent the minimum and maximum values within 1.5x the interquartile range. **d** Immunoelectron micrographs showing the localisation of giantin (arrows) in the en face view a Golgi cisterna. Signals at the edge of the cisterna are shown in blue arrowheads. Signals on the vesicles are shown by yellow arrows. The image is a representative of more than ten cells from three independent experiments. Bars, 500 nm.

observed their active movement and deformation (Fig. 5a, b). A relatively large unit had a growing partition that ended up separating one unit into two smaller units. Afterwards, a newly generated small unit frequently detached and moved away from another unit. Conversely, a unit attached another unit and finally fused to become a larger unit (Fig. 5b and Supplementary Movie 4). At higher temporal and spatial 3D resolutions by SCLIM, a partition appeared and disappeared repeatedly before two units finally fused (Fig. 5c and Supplementary Movie 5). This phenomenon was also observed in a different cell line, T47D (Supplementary Movie 6).

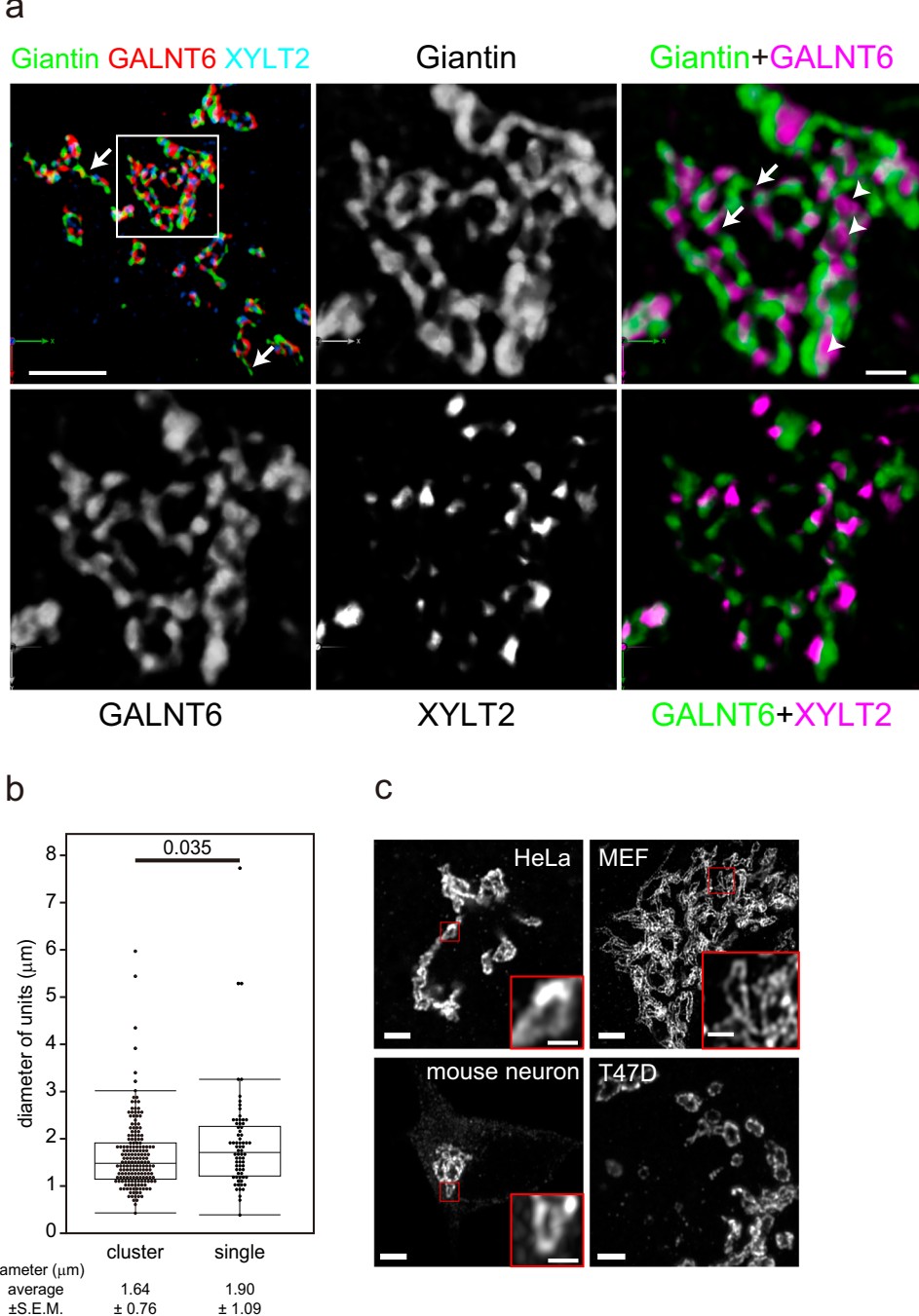

**Fig. 4 | Architecture of the Golgi in the absence of nocodazole. a** Three-colour SCLIM 3D imaging of fixed GALNT6-9xMyc + XYLT2-3xPA double KI cells. The left upper panel shows the Golgi at low magnification (Supplementary Movie 1). Other panels are magnified views of a rectangle in the left upper panel. The Golgi appears to be a cluster of Golgi units connected directly or by tubes (arrows) positive for giantin. Glycosylation enzymes are largely colocalised with giantin, whereas some enzymes are localised within the units (arrowheads). Bars, 5 μm (left upper panel), 1 μm (right upper panel) **b** Distribution of the long diameter of the Golgi units in cluster and isolated units by Airyscan. One unit is defined as circular to elongated area encircled by giantin which contains glycosylation enzymes. The number of cells: 6. The number of units: >100 (units in clusters), >70 (single units). Boxes represent 25% and 75% quartiles, lines within the box represent the median, and whiskers represent the minimum and maximum values within 1.5x the interquartile range. Statistical significance was determined by two-tailed unpaired t test. P values are depicted in the graph. **c** Golgi units whose rims are marked by giantin in different types of cells. Small red squares are magnified as insets. Bars, 2 μm (large panels), 0.5 μm (insets). Images are representative of at least three independent biological replicates. Imaging modalities and acquisition parameters of SCLIM are described in the materials and methods.

To exclude the possibility that this dynamic behaviour of the Golgi units was caused by the overexpression of *giantin*, we generated a *golgin84-Halo* knockin Caco2 cells expressing endogenous level of golgin-84 (Supplementary Fig. 6a, b). We confirmed the dynamic behaviour of the units in the *golgin-84-Halo* knockin cells (Supplementary Fig. 6c–e, Supplementary Movies 7, 8).

To determine the dynamics of glycosylation enzymes, we generated Halo-tag KI cell lines and observed the dynamics of the enzymes

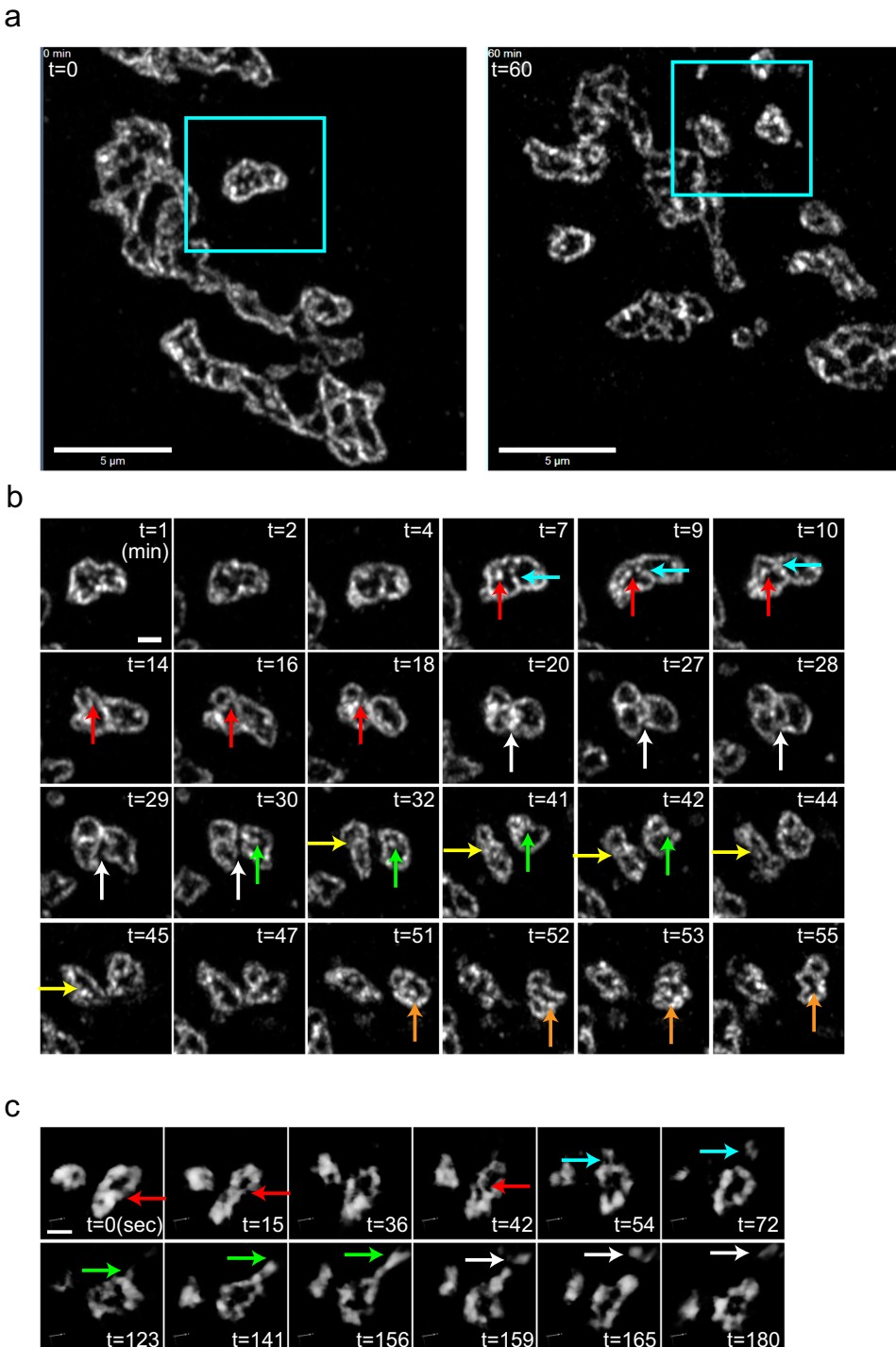

**Fig. 5 | Live imaging of the Golgi shows its dynamic movements, separation, and fusion. a** 2D Live imaging of Caco2 cells expressing *mNeonGreen-giantin* by Airyscan confocal microscopy (Supplementary Movie 4). Left: 0 min, Right: 60 min. Low magnification micrographs of the Golgi in the absence of nocodazole. Areas in the rectangles are observed for 60 minutes in **b**. Bars, 5 μm. **b** Deformation, separation, fusion, attachment and detachment of the Golgi units. Separation events are indicated by red and blue arrows (7–18 min), green arrows (30–42 min), and orange arrows (51-52 min). Fusion events are indicated by yellow arrows (32–45 min) and orange arrows (52-55 min). Detachment of two units is shown by white arrows (20–30 min). Bar, 1 μm. **c** Live imaging of Caco2 cells expressing

*mNeonGreen-golgin-84* imaged by SCLIM in 3D at higher time resolution in three dimension (Supplementary Movie 5). During fusion, a partition between units (red arrows) is present during 0-15 sec, disappears transiently at 36 sec, reappears at 42 sec, and finally disappears at 54 sec. From 54 to 72 sec, a part of the rim of the Golgi unit elongates and is cleaved and transported away (blue arrows). From 123 to 156 sec, a part of the rim elongates (green arrows). It is cleaved in the middle of the extension and is transported away (white arrows). Frame rate is 3 seconds/frame (3 seconds/volume). Bar, 1 μm. Images are representative of at least three independent biological replicates. Imaging modalities and acquisition parameters of SCLIM are described in the materials and methods.

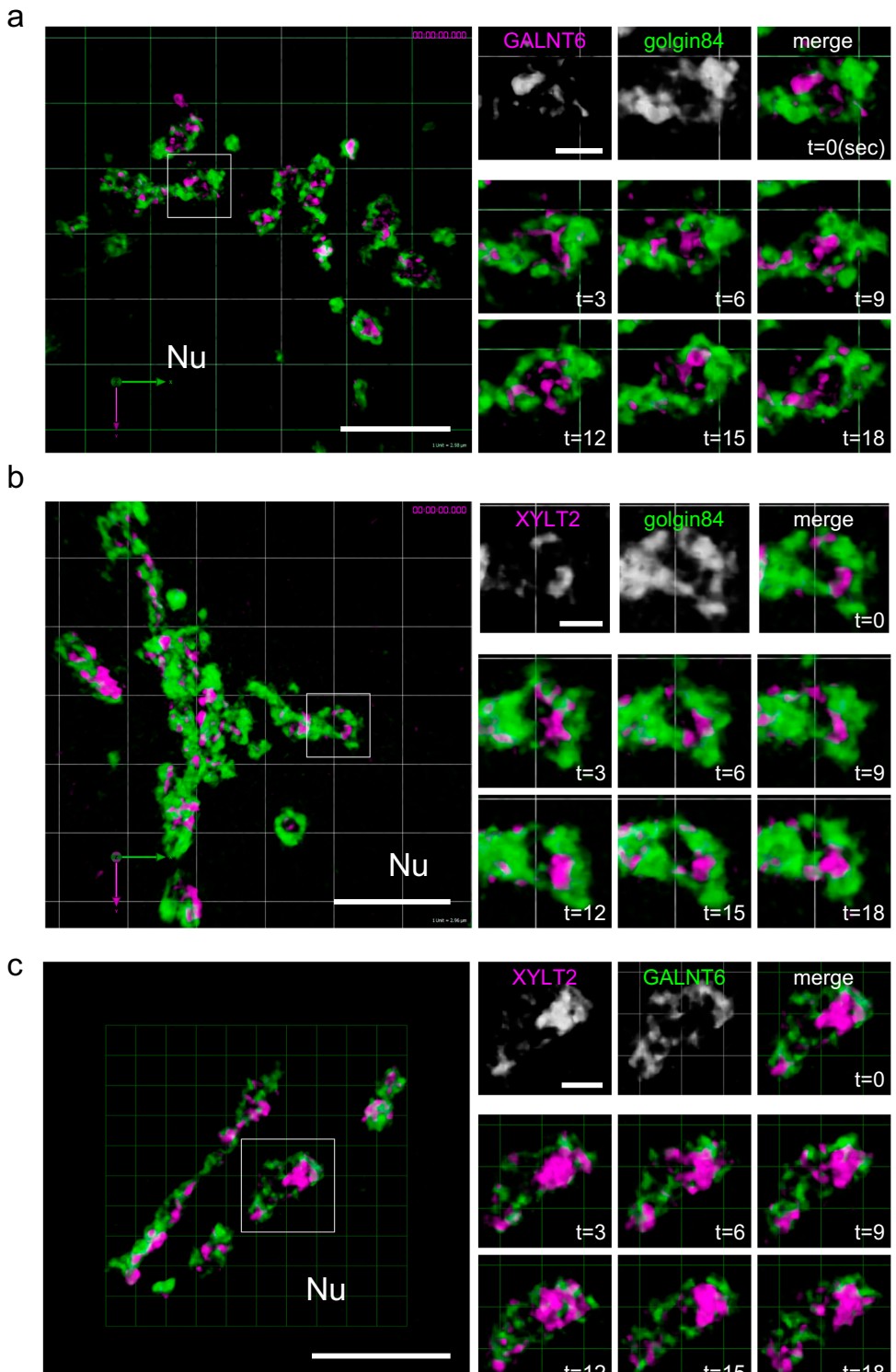

**Fig. 6 | Live imaging of the zones of glycosylation enzymes. a** 3D live imaging of a GALNT6-Halo-3xPA KI cell transfected with *mNeonGreen-golgin-84* by SCLIM (Supplementary Movie 9). Left: A low magnification micrograph of the Golgi near the nucleus (Nu). A region in the small rectangle is magnified in the right panels. Right: Top panels show GALNT6, golgin-84 and merged micrographs at 0 sec. Middle and bottom panels show the movement of GALNT6 in the Golgi unit surrounded by golgin-84 from 3 to 18 sec. A large part of the GALNT6 signal is localised near or on the rim of the unit. Frame rate is 3 seconds/frame (3 seconds/volume). Bar, 5 μm (left), 1 μm (right). **b** 3D live imaging of a XYLT2-Halo-3xPA KI cell transfected with *mNeonGreen-golgin-84* (Supplementary Movie 10). Left: a low magnification micrograph of the Golgi near the nucleus (Nu). A region in the small

rectangle is magnified in the right panels. Right: Top panels show XYLT2, golgin-84 and merged micrographs at 0 sec. Middle and bottom panels show the movement of XYLT2 in the Golgi unit surrounded by golgin-84 from 3 to 18 sec. Frame rate is 3 seconds/frame. Bar, 5 μm. **c** 3D live imaging of an XYLT2-Halo-3xPA + GALNT6-2xSNAP-9xMyc double KI cell (Supplementary Movie 13). Left: a low magnification micrograph of the Golgi near the nucleus (Nu). A region in the small rectangle is magnified in the right panels. Right: Top panels show XYLT2, GALNT6, and merged micrographs at 0 sec. Middle and bottom panels show the movements of XYLT2 (magenta) and GALNT6 (green) in the Golgi unit from 3 to 18 sec. They are localised almost mutually exclusively. Frame rate is 3 seconds/frame. Bars, 5 μm (left), 1 μm (right).

by live imaging. GALNT6 was mainly localised and moved around the rim of the units very rapidly (Fig. 6a and Supplementary Movie 9). GALNT6 did not appear to move out of the isolated unit. We observed similar rapid movements for XYLT2 and NDST1 within a unit (Fig. 6b and Supplementary Movies 10 and 11). GALNT6 exhibited similar movements in T47D cells (Supplementary Movie 12). Finally, to compare the movements of GALNT6 and XYLT2 directly, we generated and observed a double KI clone in which the XYLT2 and GALNT6 genes were tagged with Halo and SNAP tags, respectively. The zones of XYLT2 and GALNT6 moved rapidly and in mutually exclusive manner in one unit (Fig. 6c and Supplementary Movie 13).

Although we tried to determine the movement between units, their movement was too rapid for detection, even by SCLIM. We tried to observe the movement of signals at higher time resolution by observing one determined focus plane. However, we were not able to identify clusters under this condition. Therefore, to examine their movement between units, we performed fluorescence recovery after photobleaching (FRAP) analysis (Fig. 7, Supplementary Fig. 7, and Supplementary Movies 14, 15). The GALNT6 signal in an isolated unit recovered very slowly after photobleaching. When a unit in a cluster was photobleached, the signal of the bleached region recovered very slowly as well (Fig. 7a, c, Supplementary Fig. 7b, and Supplementary Movie 14). This indicates that the movement of GALNT6 is restricted within one unit regardless of whether it forms a cluster. Similar slow recovery was observed in case of giantin as well (Supplementary Fig. 7a). The signal of XYLT2 in an isolated unit recovered faintly, even less than that of GALNT6. However, the signal of a unit in a cluster recovered very rapidly (Fig. 7b, c, Supplementary Fig. 7c, and Supplementary Movie 15). Therefore, the movement of XYLT2 is restricted within one unit only when the unit is isolated, whereas XYLT2 can move rapidly between units when they are in contact. In addition, GALNT6 and XYLT2 signals appeared different during recovery after photobleaching. Although the GALNT6 signal recovered gradually, that of XYLT2 recovered as flickering puncta (Supplementary Fig. 7b, c and Supplementary Movies 14, 15).

As presented previously, the Golgi unit was surrounded by giantin and other golgins[3,4]. Among them, golgin-84, TMF1, and GCC2 showed good colocalisation with giantin at the rim of the units (Supplementary Fig. 8a). As these golgins localise at the interface when two Golgi units make contact, they might be involved in the interaction between the Golgi units. Thus, to determine their contribution to the behaviour of the Golgi units, we knocked them down in HeLa cells. Only by giantin knockdown were the Golgi units dissociated without considerable deformation of each unit (Fig. 8a, b, Supplementary Fig. 8b). We observed a similar phenotype in giantin knockout Caco2 cells (Supplementary Fig. 8c). This dissociation of units was rescued by introducing mNeonGreen-Giantin (Fig. 8a, b).

### Dissociation of Golgi units perturbs GAG synthesis

To determine the effect of giantin knockdown on the localisation of glycosylation enzymes, we measured the signals of GALNT6, MGAT1, and XYLT2 per Golgi unit in control and giantin knockdown cells. The intensity of the signal of XYLT2 distributed more broadly among Golgi units in giantin knockdown cells, namely, in some units XYLT2 signal was very weak whereas in other unis it was very high. This phenomenon was not observed in the signals of GALNT6 or MGAT1 (Fig. 8c, d and Supplementary Fig. 10a–c). FRAP analysis already showed that XYLT2 moves across the units when the units are clustered (Fig. 7b, c). This implied that if the movement of XYLT2 was blocked by dissociation of units, difference in the amount of XYLT2 signal per Golgi unit was likely to increase. To determine the physiological consequence of giantin knockdown or knockout, we measured the formation of GAG chains between control Caco2 cells and giantin-depleted cells. In giantin knockdown cells, the amount of mature GAG was reduced compared to that in control cells (Fig. 8e) which is likely to reflect the

insufficient amount of XYLT2, an essential GAG synthesising enzyme, in some Golgi units. To know whether the divergence in XYLT2 signal results from dissociation of Golgi units, we dissociated the Golgi by nocodazole. After nocodazole treatment, XYLT2 signal per unit became more diverged than the one without treatment (Supplementary Figs. 9a, b and 10d, e). Moreover, the amount of mature GAG was reduced by nocodazole treatment as well (Supplementary Fig. 9c).

Based on this result, giantin is necessary for Golgi units to attach and cluster, which allows XYLT2 to move between units to average the amount of GAG synthesising enzymes in each unit to optimise normal GAG synthesis.

## Discussion

Here, we provide structural basis to better understand the function of the Golgi complex. We showed that, in all types of cells examined, the Golgi complex is composed of a unit 1-3 μm in diameter surrounded by giantin or golgin-84 (Fig. 4 and Supplementary Fig. 6) in mNeonGreen-giantin/golgin-84 transfected cells and golgin-84-Halo knockin cells using CRISPR/Cas9 technology. Furthermore, by tag knockin, we succeeded in visualising glycosylation enzymes that accurately reflect endogenous localisation and dynamics (Supplementary Fig. 3a–c).

Since mNeonGreen-giantin can rescue Golgi fragmentation caused by giantin knockdown (Fig. 8a, b), it is functionally equivalent to nontagged giantin. In case of mNeonGreen-golgin-84 transfected cells, the rates of fusion and separation are similar to the ones of mNeonGreen-giantin transfected cells (Supplementary Table 1). In addition, in golgin-84-Halo knockin cells, the Golgi also behaved similarly to mNeonGreen-giantin transfected cells (compare Supplementary Movies 4 and 7). This clearly excludes that the observed behaviour of the Golgi units (fusion, separation, attachment, and detachment) occurs by giantin or golgin-84 overexpression, but rather reflects the endogenous behaviour of the Golgi.

From observations by super-resolution microscopies, we discovered that GALNT6, MGAT1, and XYLT2, all of which mediate early steps of glycosylation, form small domains, which we named 'zones' in a previous paper[8]. We also identified that all of these enzymes were included in one unit, confirming that the Golgi unit is a minimal functional unit of the Golgi, meaning more than morphologically defined 'mini-stack'.

We showed similar localisation of GALNT6, an O-glycosylation enzyme, and MGAT1, an N-glycosylation enzyme in the Golgi unit whereas the localisation of XYLT2, a GAG synthesising enzyme, was distinct from those of two enzymes. In addition, the XYLT2 zone is smaller than the GALNT6 zone based on the quantitative analysis of the data obtained by STORM. It was proposed that glycosylation enzymes were clustered by oligomerization or kin recognition[16,30–32]. However, as this is insufficient to explain the different behaviors of different zones, further investigation on the generation mechanism of the zone will be required.

In addition to the differences in localisation and area, GALNT6 and XYLT2 zones showed different recovery rates after photobleaching. By FRAP experiments, the GALNT6 zone moves very slowly between units even when units are connected, whereas the XYLT2 zone can move rapidly. This may be explained by the fact that the XYLT2 zone is smaller than the GALNT6 zone. If a passage between two units is larger than the XYLT2 zone and smaller than the GALNT6 zone, only the XYLT2 zone can pass through (Supplementary Fig. 11c). From electron microscopic observations, the Golgi was found to be composed of stacks connected by tubular structures side by side[33–36]. These tubules between Golgi cisternae may function as such passages that only allow the XYLT2 zone to go through.

In addition, we demonstrated the dynamics of units and zones by live imaging. We observed that the zones moved rapidly near the rim of the Golgi unit and that the GALNT6 zones and the XYLT2 zones moved largely in a mutually exclusive manner.

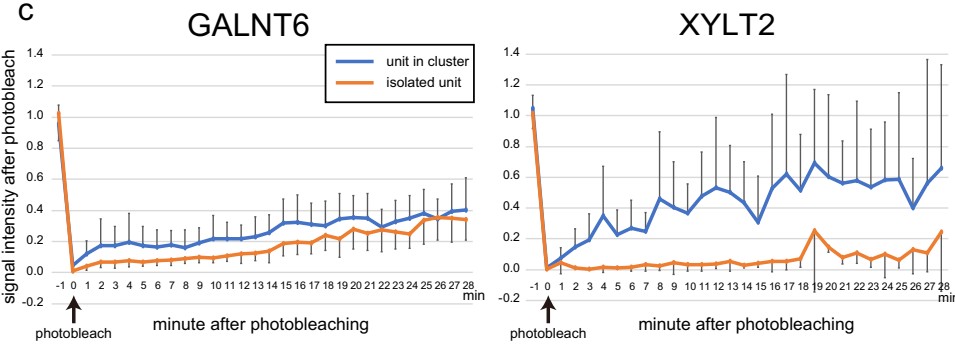

**Fig. 7 | Photobleaching of a GALNT6 and XYLT2 KI cell transfected with** ***mNeonGreen-giantin.*** **a** Photobleaching of a GALNT6-Halo-3xPA KI cell transfected with *mNeonGreen-giantin* and observed by 2D Airyscan imaging (Supplementary Movie 14). Top panels: GALNT6 and giantin signals are shown in magenta and green, respectively. A region around the green rectangle is bleached at 0 min. Bottom panels: GALNT6 signals are shown in black. A region around the red rectangle is bleached at 0 min. Arrowheads: A bleached single unit, Horizontal arrows: A bleached unit in cluster. Vertical arrows: a unit in the nonbleached region. Bar, 1 μm. **b** Photobleaching of an XYLT2-Halo-3xPA KI cell transfected with *mNeonGreen-giantin* and observed by 2D Airyscan imaging (Supplementary Movie 15). Top panels: XYLT2 and giantin signals are shown in magenta and green,

respectively. A region around the green rectangle is bleached at 0 min. Bottom panels: XYLT2 signals are shown in black. A region around the red rectangle is bleached at 0 min. Arrowheads: A bleached single unit, Horizontal arrows: A bleached unit in cluster. Vertical arrows: a unit in the nonbleached region. Bar, 1 μm. **c** Time-dependent FRAP measurements on GALNT6-Halo and XYLT2-Halo and quantification of FRAP recovery as a function of time until 28 min after photobleaching by 2D Airyscan imaging. Orange and blue lines indicate averaged Halo signals in an isolated single unit and a unit in a cluster, respectively. Graphs of signals from individual experiment are shown in Supplementary Fig. 7b, c. N = 5 for each. Error bars represent SEM. In **a**–**c**, images are representative of at least three independent biological replicates.

Though these early glycosylation enzymes form zones around the rim of a single cis or medial cisterna, zones of late glycosylation enzymes, particularly β4GalT1, occupy a more central position within a trans cisterna. β4GalT1 is responsible for the synthesis of N-linked and O-linked oligosaccharides in many glycoproteins as well as the

carbohydrate moieties of glycolipids. It might be convenient for this enzyme to localize at the center of the trans cisterna to modify a number of substrates that come from various positions in cis-medial cisternae. In contrast, as GAG synthesising late-stage enzyme NDST1 is responsible only for the synthesis of GAG, it is more efficient for it to

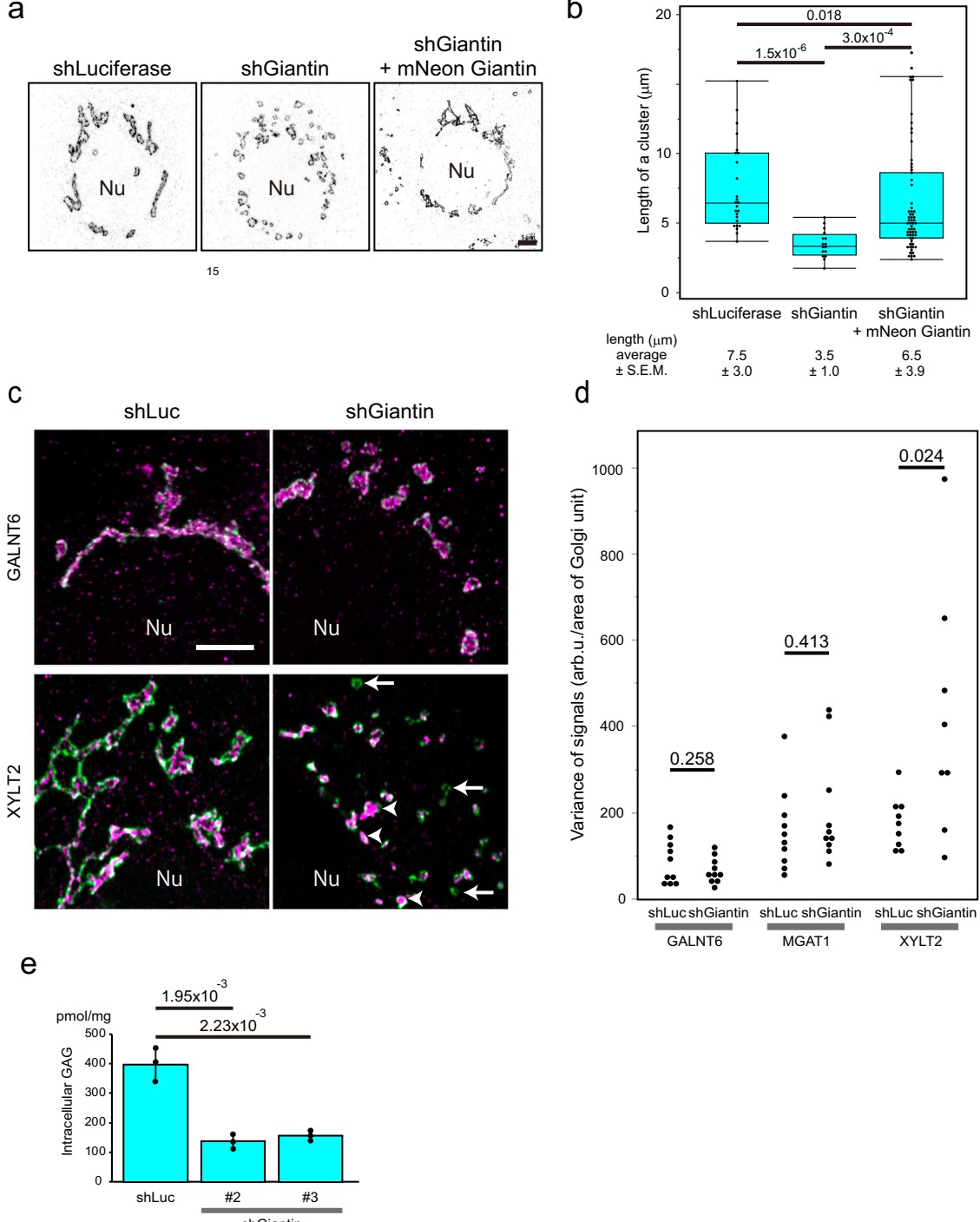

**Fig. 8 | Giantin is necessary for clustering Golgi units. a** Scattered Golgi units imaged by Airyscan in 2D in *giantin* knockout cells (middle) are reassembled by expressing *mNeonGreen-giantin* (right). The rim of the unit is stained by TMF1 in black. Nu: nucleus. Bar, 5 μm. **b** A graph showing the rescue experiment. The number of units per cluster of the Golgi is shown. Number of cells: 6 (shLuciferase), 7 (shGiantin), 8 (shGiantin+mNeonGreen-giantin). Number of units: 25 (shLuciferase), 16 (shGiantin), 63 (shGiantin+mNeonGreen-giantin). P values are depicted in the graph. Boxes represent 25% and 75% quartiles, lines within the box represent the median, and whiskers represent the minimum and maximum values within 1.5 x the interquartile range. The statistical significance was determined by the two tailed unpaired t-test. P values are depicted above the plots in the graph. **c** Representative images of sh*Luciferase*- and sh*Giantin*-treated cells by Airyscan in 2D. GALNT6 KI (upper panels) and XYLT2 KI (lower panels) Caco2 cells are stained with TMF1 (green) and GALNT6 (magenta) (upper panels) or with TMF1 (green) and XYLT2

(magenta) (lower panels). In the right lower panel, arrows and arrowheads indicate the Golgi units that contain very low and high amounts of XYLT2, respectively. In a and c, images are representative of at least three independent biological replicates. Bar, 5 μm. **d** Effects of *giantin* knockdown on the distribution of signal intensity of glycosylation enzymes per Golgi unit in Caco2 cells. The distribution of signal intensity is shown by variance of signals in each cell in Supplementary Fig. 10. The statistical significance of the variance was determined by the two tailed unpaired t-test. P values are depicted in the graph. **e** *Giantin* knockdown Caco2 cells display reduced amount of GAG (heparan sulphate) trapped within the cells. Lentivirus encoding two different shRNA (#2 and 3) are used for *giantin* knockdown. Number of samples: 3 (shLuciferase), 3 (shGiantin #2), 3 (shGiantin #3). The statistical significance was determined by the two tailed unpaired t-test. P values are depicted above the plots in the graph. Error bars represent SD.

localise near the rim just for recycling back to the previous cisterna by COPI vesicle.

There are several advantages to consider the tiny Golgi unit as a functional unit when we explain the function of the Golgi complex. The Golgi complex must perform different types of glycosylation reactions and other modifications very rapidly. If all sets of rapidly moving enzymes localise in one tiny unit, the chance of substrates entering the zone of their target enzymes is greatly increased even within a short period of time.

The paper from Fourriere et al.[29] reports that many of the Golgi mini-stacks formed in nocodazole treated HeLa cells are not fully functional for secretion. To distinguish the effect of Golgi dispersion from microtubule depolymerisation, they expressed a dominant-negative dynein construct, p150-CC1, to disperse the Golgi without microtubule depolymerisation (Fig. 3 in Fourriere et al.[29]). They observed a normal transport of TNF1 in Golgi dispersed cells by p150-CC1 overexpression. However, when they further treated nocodazole to p150-CC1 expressed cells, they observed blockade of TNF1 transport. Our *giantin* knockdown or knockout cells seems similar to the case of p150-CC1 expressing cells because microtubules are not depolymerised. Thus, giantin-negative ministacks by giantin depletion are likely to be competent for cargo transport to the plasma membrane. On the other hand, we observe greater divergence in the amount of XYLT2 among the isolated Golgi units in the absence of giantin. This indicates that cargos for the plasma membrane can go through the Golgi units generated by giantin depletion, although some of them acquire insufficient glycosylation of GAG because of insufficient amount of its enzymes.

By live imaging, a large unit frequently develops a partition to make new small units. This may indicate that some mechanisms which sense the area of the unit is likely to exist. Our live imaging also demonstrated the dynamic nature of the units. They frequently exert attachment, detachment, fusion and separation in Caco2 and T47D cells (Supplementary Fig. 11b). In HeLa cells, units are usually tightly clustered near the nucleus, which might have made it difficult to observe the dynamics of units thus far.

When the units were dissociated by *giantin* knockdown, deviated amounts of XYLT2 per unit and reduced amount of GAG were observed. Our paper provides a basis to explain this relationship. Namely, giantin deficiency causes units to dissociate, which will prevent interunit movements of GAG-synthesising enzymes. As the amounts of enzymes between units will deviate as a result, some units will contain insufficient amounts of enzymes for GAG synthesis. Previously, *giantin* knockout mice or rats were shown to develop defects in GAG or proteoglycan-rich tissues, such as bones or cartilages[37,38]. In humans, defects in several membrane tethering proteins cause diseases in whose bones and cartilages are affected[39]. Our observation is instructive to explain the pathogenesis of these diseases.

Golgi fragmentation has been previously observed by depletion of giantin[40–42]. In one of the previous reports, the cisternal length of the Golgi was found to be similar between control and *giantin* KO cells[42], which apparently contradicts our report. This contradiction can be explained by the average length of the cisternae. Namely, in these previous reports, the average length of the Golgi is around 1 μm which corresponds to the diameter of the Golgi units in our paper (Fig. 4b). Therefore, they likely measured the diameter of Golgi units rather than the diameter of Golgi ribbon composed of a number of units. Other proteins such as GM130 and GRASPs[43–46] have also been known to link the Golgi units. This indicate that the redundancy of these proteins, particularly between GRASPs and giantin, for linking Golgi units to make sure that the connection will not be broken easily.

Golgi fragmentation has been reported in Alzheimer's and other neurological diseases[47,48]. However, the relationship of Golgi fragmentation and pathogenesis of these diseases has been elusive so far. As defects in synthesis and modification of GAG has been reported in Alzheimer's disease[49], Golgi fragmentation may contribute to the pathogenesis through regulation of GAG synthesis and/or modifications.

In summary, we determined the morphological characteristics of the Golgi and its glycosylation enzymes, which can well explain its various functions. Although we showed that giantin is important for the attachment of Golgi units, most of the molecular mechanisms underlying Golgi dynamics remain unclear. Answering these questions will lead to further knowledge on this long-known yet enigmatic organelle.

## Methods
### Cell culture
Caco2 cells, HeLa cells, and mouse embryonic fibroblasts (MEFs) were grown at 37 °C with 5% CO2 in DMEM with 10% (HeLa, MEFs) or 15% (Caco2) FBS. T47D cells were grown at 37 °C with 5% $CO_2$ in DMEM/Ham's F12 with 5% FBS. Primary neurons were grown at 37 °C with 5% $CO_2$ in neurobasal medium containing B-27 and GlutaMAX. For nocodazole treatment of Caco2 and HeLa cells, we first placed the culture dish on ice for 10 min, followed by culturing in nocodazole (5 mg/ml)-containing medium for 1-2 hr (Caco2) and 6 hr (HeLa) at 37 °C with 5% $CO_2$. Transfection of plasmid DNA in HeLa cells was carried out using FuGENE6 as previously described[50,51]. Transfection of plasmid DNA (5 μg/electroporation) in Caco2 and T47D cells was carried out by electroporation as described in the next section. For the SDC2 rush assay, a rush vector for SDC2 tagged with a 6x flag was electroporated into Caco2 cells. Electroporated Caco2 cells were plated onto collagen-coated coverslips and used the next day. For the assay, cells were incubated in prewarmed DMEM containing 40 μM biotin and 0.1 mM cycloheximide. The assay was stopped by incubating the coverslips in PBS containing 3% PFA.

### Generation of knockin clones
Transfection of plasmid DNA in Caco2 and T47D cells was carried out using a NEPA21 electroporation system (Nepa Gene). For Caco2 and T47D cells, $2 \times 10^6$ cells were washed twice with Opti-MEM (Gibco # 11058021) and then resuspended in 100 μl Opti-MEM with 3.75 μg of template DNA and 1.25 μg of pX330 with a target sgRNA sequence (written in the next section) in an electroporation cuvette with a 2-mm gap (Nepa Gene # EC-002S).

Caco2 cells were electroporated with a poring pulse of 170 V, 7.5 ms pulse length, 50 ms pulse interval, 2 pulses, with decay rate of 10% and + polarity; consecutively with a transfer pulse of 20 V, 50 ms pulse length, 50 ms pulse interval, 5 pulses, with a decay rate of 40% and ± polarity.

T47D cells were electroporated with a poring pulse of 125 V, 5 ms pulse length, 50 ms pulse interval, 2 pulses, with decay rate of 10% and + polarity; consecutively with a transfer pulse of 20 V, 50 ms pulse length, 50 ms pulse interval, 5 pulses, with a decay rate of 40% and ± polarity.

After electroporation, Caco2 and T47D cells were cultured in various drugs (puromycin, hygromycin, blasticidin S) to select clones. Hygromycin, blasticidin S, and puromycin (all Gibco) were used at the following concentrations: puromycin 5 μg/ml (Caco2) or 0.5 μg/ml (T47D), hygromycin 60-100 μg/ml (Caco2) or 15 μg/ml (T47D), and blasticidin S 5 μg/ml (Caco2). When drug-resistant colonies grew large enough (usually from two to four weeks after electroporation), they were picked up by pipetman under a stereo microscope. The colonies were trypsinized and cultured in the same medium in 96-well plates (flat bottom) (Corning). They were expanded for screening by immunofluorescence and PCR.

### DNA construction for CRISPR knockin
For CRISPR–Cas9 gene editing, all guide RNAs were designed using the CRISPRdirect website (https://crispr.dbcls.jp/). The following sgRNA

target sequences were cloned into the BbsI site of a pX330 vector (Addgene #42230).

*β4GalT1*: 5′-GGGACACCGAGCTAGCGTTTTGG-3′
*GALNT6*: 5′-CCAGTGACCCCCATCAGTTGTGG-3′
*MGAT1*: 5′-AGGAGGGGCCCAGGAAGGACAGG-3′
*NDST1*: 5′-GAGGTCCTCTCGTAGCCAAGTGG-3′
*XYLT2*: 5′-TCAAAGCAGACGGGCGACTCAGG-3′
*GOLGA5*: 5′-AACCAATCACTGCAACAACTGGG-3′

To generate homologous recombination (HR) plasmids for 9xMyc tagging, a backbone plasmid containing tag and drug resistance sequences was constructed. A 9xMyc tag sequence, which was obtained from the pMK76 plasmid (provided by the National Bio-Resource Project (NBRP) of AMED, Japan), was inserted into the PstI site of a pBluescript II KS (+) vector (Agilent Technologies) in two different directions. A puromycin resistance sequence with the SV40 promoter and polyadenylation signal (SV40-puro-polyA) was inserted downstream of a 9xMyc tag. The SV40 promoter and puromycin resistance sequences were flanked by loxP sites. This backbone vector was referred to as pBS-9xMyc-puro-loxP #1 or #2. Except for the pBS-3xPA-hygro-loxP vectors (described below), the other backbone vectors containing different tags and drug resistance sequences were generated using the same strategy. All backbone vectors generated in this work are listed in Supplementary Table 2.

For the GALNT6-9xMyc-puro HR plasmid, DNA fragments of the 5′ homology arm (HA) and 3′ HA were amplified from Caco2 genomic DNA by PCR using the following primers.

5′ HA with XhoI forward: 5′-GGCTCGAGGCATCGTACGCGT ACGTGTTTGGAGGAACCTAACCTGGAGGCTGCAGG-3′

5′ HA with EcoRI reverse: 5′-GGGAATTCGACAAAGAGCCACA ACTGATGGGG-3′

3′ HA with SpeI forward: 5′-GGACTAGTGACCCAGATCATCCCCAG AGAGAG-3′

3′ HA with NotI reverse: 5′-GGGCGGCCGCGCATCGTACGCGTA CGTGTTTGGAGCTGCGACTACAGGCATAAGCCAC-3′

The 5′ HA and 3′ HA were inserted into the XhoI-EcoRI and SpeI-NotI sites of the pBS-9xMyc-puro #2 vector, respectively.

For the NDST1-9xMyc-puro HR plasmid, 5′ HA and 3′ HA were amplified from Caco2 genomic DNA by PCR using the following primers.

5′ HA with NotI forward: 5′-GGGCGGCCGCGCATGGTGGGTTT TTTGTTTTTTT-3′

5′ HA with SpeI reverse: 5′-GGACTAGTCCTGGTGTTCTG-GAGGTCCTCTCG-3′

3′ HA with EcoRI forward: 5′-GGGAATTCGGATCTGCAAGCACCT CGGAGCAC-3′

3′ HA with ApaI reverse: 5′-AAGGGCCCTCCGGGGACCCAGCTG GCCCCTTG-3′

The 5′ HA and 3′ HA were inserted into the NotI-SpeI and EcoRI-ApaI sites of the pBS-9xMyc-puro #1 vector, respectively.

To generate HR plasmids for 3xPA tagging, the XYLT2-3xPA-hygro HR plasmid was constructed first. A 5′ HA was amplified from Caco2 genomic DNA by PCR using the following primers and was inserted into the NotI-XbaI sites of pBluescript II KS (+).

5′ HA with NotI forward: 5′-GCGGCCGCGCATCGTACGCGTACGT GTTTGGTTGAACCCACTGCTTTACTGCATC-3′

5′ HA with XbaI reverse: 5′-TCTAGAGCGGAGTCGCCCGTCTG CTTTGAC-3′

A 3xPA tag sequence was cut out from the pCD-PAx3 vector (a kind gift from Dr. Jun-ichi Takagi, Institute for Protein Research, Osaka University, Japan) using XbaI and PmeI and was inserted downstream of the XYLT2 5′ HA using XbaI-EcoRV sites. A hygromycin B resistance sequence with an SV40 promoter and polyA signal (SV40-hygro-polyA) was amplified from the pcDNA3.1-Hygro vector (Invitrogen) using the following primers.

Hygro with HindIII forward: 5′-GGGAAGCTTTGTGTCAGT-TAGGGTGTGGAA-3′

Hygro with BamHI, SpeI and ApaI reverse: 5′-GGGGGGCCCAC-TAGTCCGGATCCATAAGATACATTGATGAGTTTGGAC-3′

This fragment was inserted downstream of a 3xPA tag using HindIII-ApaI sites.

A 3′ HA was amplified from Caco2 genomic DNA by PCR using the following primers and was inserted downstream of an SV40 polyA signal using BamHI-SpeI sites.

3′ HA with BamHI forward: 5′-GGGGATCCGAAATTGCACCTTA-CAGACAGTGG-3′

3′ HA with SpeI reverse: 5′-GGACTAGTGCATCGTACGCGTACGT GTTTGGATCACCTGTTCTTCGGGCCTTAG-3′

The fragment from the 3xPA to SV40 polyA signal was amplified from the XYLT2-3xPA-hygro plasmid by PCR and cloned into the BamHI site of a pBluescript II KS (+) vector in two different directions. Two loxP sites were inserted upstream of an SV40 promoter and downstream of a hygromycin B resistance sequence by quick-change mutagenesis. This backbone vector was referred to as pBS-3xPA-hygro-loxP #1 or #2.

For the β4GalT1- and MGAT1-3xPA-hygro HR plasmids, DNA fragments of 5′ HA and 3′ HA were amplified from Caco2 genomic DNA by PCR using the following primers.

*β4GalT1* 5′ HA with NotI forward: 5′-GGGCGGCCGCGCATCG-TACGCGTACGTGTTTGGCAACAGTCTTGCAGCCGGGGCTCTC-3′

*β4GalT1* 5′ HA with SpeI reverse: 5′-GGACTAGTGCTCGGTGTCC CGATGTCCACTGTG-3′

*β4GalT1* 3′ HA with EcoRV forward: 5′-GGGATATCGATAAGAGAC CTGAAATTAGCCAGG-3′

*β4GalT1* 3′ HA with XhoI reverse: 5′-GGCTCGAGGCATCGTACGC GTACGTGTTTGGGTTGCATACCACTGAAAGAAGGGGG-3′

*MGAT1* 5′ HA with NotI forward: 5′-GCGGCCGCAGCTCTGGGCTGA GCTGGAGCCC-3′

*MGAT1* 5′ HA with SpeI reverse: 5′-ACTAGTATTCCAGCTAGGATCA TAGCCCTC-3′

*MGAT1* 3′ HA with EcoRV forward: 5′-GATATCCACATCATGA GCTGAGGTGGGAC-3′

*MGAT1* 3′ HA with XhoI reverse: 5′-CTCGAGCAGACAGGAGAGAA GGGCAGGAG-3′

The 5′ HA and 3′ HA were inserted into the NotI-SpeI and EcoRV-XhoI sites of the pBS-3xPA-hygro-loxP #1 vector, respectively.

For the GALNT6-3xPA-hygro HR plasmid, DNA fragments of 5′ HA and 3′ HA were amplified from Caco2 genomic DNA by PCR using the following primers.

5′ HA with XhoI forward: 5′- GGCTCGAGGCATCGTACGCGTACG TGTTTGGAGGAACCTAACCTGGAGGCTGCAGG-3′

5′ HA with EcoRV reverse: 5′-GGGATATCGACAAAGAGCCACAAC TGATGGGG-3′

3′ HA with SpeI forward: 5′-GGACTAGTGACCCAGATCATCCCCAG AGAGAG-3′

3′ HA with NotI reverse: 5′- GGGCGGCCGCGCATCGTACGCGTAC GTGTTTGGAGCTGCGACTACAGGCATAAGCCAC-3′

The 5′ HA and 3′ HA were inserted into the XhoI-EcoRV and SpeI-NotI sites of the pBS-3xPA-hygro-loxP #2 vector, respectively.

For the GALNT6-Halo-3xPA-puro HR plasmid, a 9xMyc-puro sequence in the GALNT6-9xMyc-puro plasmid was replaced with a Halo-3xPA-puro sequence, which was cut out from the pBS-Halo-3xPA-puro #2 backbone vector, using EcoRI and SpeI. Similarly, a 3xPA-hygro sequence in GALNT6-3xPA-hygro was replaced with a 2xSNAP-9xMyc-hygro sequence, which was cut from the pBS-2xSNAP-9xMyc-hygro #2 backbone vector by EcoRV and SpeI digestion, to generate the GALNT6-2xSNAP-9xMyc-hygro plasmid.

To generate the XYLT2-Halo-3xPA-puro and XYLT2-3xV5-blast HR plasmids, the XYLT2-APEX2-3xPA-puro HR plasmid was constructed

first. DNA fragments of 5′ HA and 3′ HA were amplified from Caco2 genomic DNA by PCR using the following primers.

5′ HA with NotI forward: 5′-GCGGCCGCGCATCGTACGCGTACG TGTTTGGTTGAACCCACTGCTTTACTGCATC-3′

5′ HA with XbaI reverse: 5′-TCTAGAGCGGAGTCGCCCGTCTGC TTTGAC-3′

3′ HA with EcoRI forward: 5′-GGGAATTCGAAATTGCACCTTACAG ACAGTGG-3′

3′ HA with SalI reverse: 5′-GGGTCGACGCATCGTACGCGTACGT GTTTGGATCACCTGTTCTTCGGGCCTTAG-3′

The 5′ HA and 3′ HA were inserted into the NotI-XbaI and EcoRI-SalI sites of the pBS-APEX2-3xPA-puro #1 vector, respectively. An APEX2-3xPA-puro sequence in the XYLT2-APEX2-3xPA-puro plasmid was replaced with a Halo-3xPA-puro or a 3xV5-blast, which was cut out from the pBS-Halo-3xPA-puro #1 or the pBS-3xV5-blast #1 backbone vector, using XbaI and EcoRI.

For the GOLGA5-Halo-3xPA-puro HR plasmid, 5′ HA and 3′ HA were amplified from Caco2 genomic DNA by PCR using the following primers.

5′ HA with NotI forward: 5′-GGGCGGCCGCACATCAGCCTCTGA AACGACAC-3′

5′ HA with SpeI reverse: 5′-GGACTAGTTTTGCCATATGGTTGGT CGTGGTG-3′

3′ HA with EcoRV forward: 5′-GGGATATCCTAGACTTGGGAT CTGCAAGAAGG-3′

3′ HA with ApaI reverse: 5′-AAAGGGCCCACTAAAGTCCGTCTGGA AAGGTC-3′

The 5′ HA and 3′ HA were inserted into the NotI-SpeI and EcoRV-ApaI sites of the pBS-Halo-3xPA-puro #1 vector, respectively.

## Production of lentivirus for knockdown

The following lentiviral plasmids were transfected with envelope and packaging plasmids (pMD2. G and psPAX2, respectively) into 293FT cells using TransIT-LT1. The culture medium was collected, filtered through a 0.45 μm syringe filter, and used for transduction. Lentiviral pLKO1 (a gift from Bob Weinberg; Addgene plasmid #8453) puro plasmids for knockdown experiments were ligated with oligonucleotides against human *giantin* (shRNA#2 target sequence 5′- ACT TCATGCGAAGGCCAAA −3′, shRNA#3 5′- CTGGAGTAGAAT TGAAATCAA −3′) and *golgin-84* (5′- CCATTCAAAGAGTGATCGAAT −3′). Lentiviral shRNA plasmids for *GCC2* (VectorBuilder #VB220524-1029cty) and *TMF1* (VectorBuilder #VB220524-1019xzb) were purchased from VectorBuilder Japan. The plasmids were transfected with envelope and packaging plasmids into 293FT cells using TransIT-LT1. Culture medium containing lentiviral particles was filtered through a 0.45 μm syringe filter and concentrated 80-fold in PBS as previously described[50,51]. Cells were infected with lentivirus containing medium diluted twofold in the presence of 10 μg/ml polybrene for 72 h at 37 °C. After infection, the culture medium was replaced with virus-free medium with 5 μm/cc blastocidin S.

## Western blotting

The protocols for SDS–PAGE and immunoblotting or lectin blotting[52] were described previously. The following primary antibodies or biotinylated lectins were used: anti-β4GalT1 (GeneTex), anti-Myc (MBL Life Science), anti-PA rat monoclonal antibody[14], PNA (EY LABORATORIES), and VVA (EY LABORATORIES). HRP-labelled donkey anti-rat and anti-rabbit antibodies (Jackson ImmunoResearch Laboratories) and HRP-conjugated streptavidin (Jackson ImmunoResearch Laboratories) were used as secondary antibodies. An Immobilon Western Chemiluminescent HRP Substrate kit (Millipore) was used for signal detection. Chemiluminescent images were obtained using a ChemiDoc Touch MP system (Bio-Rad).

## Immunofluorescence analysis

Cells grown on collagen-coated coverslips were fixed with 3% paraformaldehyde (PFA) (Fuji Film) in PBS for 10–15 min at room temperature (RT). After fixation, coverslips were incubated with blocking buffer (PBS containing 5% normal donkey serum and 0.25% saponin) for 5 min and stained with primary antibodies in blocking buffer for 1 h at 37 °C. The coverslips were washed with PBS three times and were blocked in blocking buffer as described. Then, the coverslips were incubated with fluorescently labelled secondary antibodies in blocking buffer for 1 h at RT. For long-term storage, some samples were fixed with 3% PFA and 0.1% glutaraldehyde in PBS for 10–15 min at room temperature.

## Antibodies for immunofluorescence

The following primary antibodies were used for immunofluorescence: anti-PA rat monoclonal antibody[14], anti-Myc rabbit polyclonal antibody (MBL Life Science #562), anti-V5 mouse monoclonal antibody (ThermoFisher #R960-25), anti-GALNT6 mouse monoclonal antibody and rabbit polyclonal antibody[53], anti-β4GalT1 mouse monoclonal antibody[20], anti-C1GalT1 mouse monoclonal antibody (Millipore #MABT1555), anti-Giantin (N terminal region) mouse monoclonal antibody (Abcam #ab37266), anti-Giantin (C terminal region) rabbit polyclonal antibody (BioLegend #924301), anti-TMF1 rabbit polyclonal antibody (Sigma #HPA008729), anti-golgin-84 rabbit polyclonal antibody (Sigma #HPA000992), anti-GCC2 rabbit polyclonal antibody (Sigma #HPA035849), anti-CASP rabbit monoclonal antibody (Abcam #ab182216), anti-GMAP210 rabbit polyclonal antibody (Proteintech #26456-1-AP), anti-golgin-97 polyclonal antibody[52], anti-golgin-245 mouse monoclonal antibody (BD Biosciences # 611281), anti-GRASP55 rabbit polyclonal antibody (Novus Biologicals #NBP2-38244), anti-GRASP65 polyclonal antibody[54], anti-GM130 mouse monoclonal antibody (BD Biosciences #610822), anti-DYKDDDDK mouse monoclonal antibody (clone #1E6, Fuji Film). GALNT7 rabbit polyclonal antibody was raised in rabbits by injecting peptide encoding 40-56 amino acid residues of human GALNT7. The following secondary antibodies and reagents were used: donkey anti-mouse IgG (H + L) highly cross-adsorbed secondary antibody, Alexa Fluor 488 (Invitrogen, #A-21202), donkey anti-rabbit IgG (H + L) highly cross-adsorbed secondary antibody, Alexa Fluor 568 (Invitrogen, #A-10042), donkey anti-rat IgG (H + L) highly cross-adsorbed secondary antibody, Cy5 (Jackson Immuno Research, #712-175-153), DAPI (4′, 6-Diamidine-2′-phenylindole dihydrochloride, Sigma Aldrich #10236276001).

## Immunoelectron microscopy

Cells were fixed with 3% PFA and 0.1% glutaraldehyde in PBS, pH 7.4, for 30 min at RT and washed with PBS. Free aldehyde was quenched with 5-10 mg/ml NaBH4 in PBS containing 0.25% saponin for 30 min at RT. In cases of staining NDST1 and β4GalT1, cells were fixed with PLP fixative (10 mM NaIO4, 75 mM Lysine, 4% PFA in 37.5 mM phosphate buffer pH 7.2) for 2 h at RT. After fixation, the cells were permeabilized in blocking solution (5% normal goat serum and 0.25% saponin in PBS) and incubated with rat monoclonal antibody against PA tag or giantin (BioLegend) in combination with mouse monoclonal antibody against β4GalT1[20], GM130 (BD Biosciences) or rabbit antisera against Myc (MBL Life Science) in blocking solution for 1-2 h at 37 °C. After washed with PBS three times, they were incubated with secondary antibodies (goat anti-rabbit or anti-rat Alexa Fluor 488-labelled FluoroNanogold (Nanoprobes) in combination with biotin-labelled anti-mouse or anti-rabbit (Vectastain ABC kit Elite from Vector Laboratories) in blocking solution containing 1.5% normal horse (for rabbit primary antibody) or goat serum (for rat primary antibody) and 0.25% saponin in PBS for 1 h at RT. After incubation with secondary antibodies, the samples were incubated with PBS containing ABC complex (Vector Laboratories) for 30 min at RT. After the samples were washed with PBS and TBS (pH 7.4)

once for each, they were incubated in 3,3' diaminobenzidine (DAB) solution in TBS for 5 min, followed by DAB solution 4 ml in TBS with 0.5% $H_2O_2$ for 20 min. After washed with TBS twice, the samples were postfixed with 3% PFA and 0.1% glutaraldehyde in PBS, pH 7.4, for 10 min at RT. Then, the samples were washed with PBS three times and with distilled water twice. The samples were silver enhanced with HQ silver enhancement solution (Nanoprobes) for 3–7 min at RT and then extensively washed with distilled water and incubated with selenium toner for 7 min at RT. After washed with distilled water, the cells were fixed with 2.5% glutaraldehyde in Hepes buffer (30 mM Hepes, 100 mM NaCl, 2 mM CaCl2, pH 7.4) 2 h overnight, washed with Hepes buffer and postfixed in 1% osmium tetroxide and 1.5% potassium ferrocyanide in the same buffer for 40 min on ice. The fixed cells were washed with distilled water, stained with 4% uranyl acetate in distilled water for 2 h overnight, dehydrated, and embedded as previously described[55].

### Airyscan imaging
Immunofluorescence images of Caco2, HeLa, T47D, MEFs, or primary neuronal cells were obtained using a Zeiss LSM880 Airyscan confocal microscope with a PlanApo 63x objective lens (NA 1.4; Carl Zeiss). Approximately 5 to 10 slices of the z-step image were taken at 0.5 μm intervals. Each raw image was processed by Airyscan processing to obtain a superresolution image, and z-stack images were generated by the maximum intensity projection using ZEN software (Carl Zeiss).

### Time-lapse and FRAP imaging by Airyscan
To perform live imaging with the Golgi units, an expression vector for fluorescent-tagged *giantin* or *golgin-84* was constructed. The plasmid containing a full-length cDNA of rat *giantin* was a kind gift from Dr. Ayano Satoh (Okayama University). A full-length cDNA of human *golgin-84* was amplified from Marathon-Ready cDNA (Clontech #7400-1). The C-terminal partial sequence of human *giantin* in the *mNeonGreen-Giantin* plasmid (Addgene #98880) was replaced with a full-length sequence of rat *giantin* or human *golgin-84*. The *mNeonGreen-giantin* or *mNeonGreen-golgin-84* plasmid was electroporated into GALNT6-Halo or XYLT2-Halo KI Caco2 cells using a NEPA21 type II electroporator (NEPAGENE). Cells were seeded on a 35-mm glass-bottom dish (Matsunami) and cultured in DMEM containing 15% FCS for 18 h at 37 °C. Live cell imaging was performed using a Zeiss LSM880 Airyscan confocal microscope with a PlanApo 63 x objective lens. Time-lapse images were acquired every 1 min for 30 or 60 min at 37 °C and processed as described above.

To analyse fluorescence recovery after photobleaching (FRAP), the cells were incubated in medium containing 1 mM Janelia Fluor 549 ligand (Promega) and 0.1 mg/ml cycloheximide for 1 h at 37 °C. Time-lapse Airyscan images were obtained as described above. Two minutes after the start of imaging, GALNT6-Halo or XYLT2-Halo signals in the clustered and single Golgi units were quenched by scanning with a 561 nm laser 10 times. Fluorescence recovery was subsequently imaged every 1 min for 28 min.

### Time-lapse imaging by SCLIM
SCLIM was developed by combining an Olympus model IX-73 inverted fluorescence microscope with a UPlanXApo 100× 1.4–numerical aperture oil objective lens (Olympus), a high-speed and high signal-to-noise ratio spinning-disk confocal scanner (Yokogawa Electric), a custom-made spectroscopic unit, image intensifiers (Hamamatsu Photonics) equipped with a custom-made cooling system, magnification lens system to achieve × 266.7 final magnification, and electron-multiplying charge-coupled device cameras (Hamamatsu Photonics)[6,9]. Image acquisition was executed by custom-made software (Yokogawa Electric). For 3D observations, we collected 31 optical sections spaced 0.1 μm apart (z-range = 3 μm) apart in stacks by oscillating the objective lens vertically with a custom-made piezo actuator. For 4D observations, we collected 31 or 36 optical sections

spaced 0.2 μm apart (z-range= 6–7 μm). The x and y axes spatial resolution of SCLIM is 180 nm, 180 nm, 240 nm for green, red, and infrared fluorescence channels, respectively. Z-stack images were reconstructed to 3D images and deconvolved by using theoretical point spread functions with Volocity (Quorum Technologies). In Volocity, we first applied the images with a median filter to remove noise and then deconvolve the images by an inverse filter method ("Fast restoration" function in Volocity). The 3D images are displayed using "3D opacity" mode. Calculation of the Pearson correlation coefficients between the signal intensities of each voxel on the Golgi was performed after three-dimensionally segmenting individual Golgi as ROI with Volocity. Thresholds for the calculation of correlation coefficients were automatically determined.

For live imaging, an *mNeonGreen-giantin* or *mNeonGreen-golgin-84* expression plasmid was electroporated into Caco2 or T47D cells. Electroporated cells were plated onto collagen-coated coverslips and used on the next day. For live imaging, cells were incubated in medium containing 0.4 μM Janelia Fluor 549 Halo ligand (Promega) (and 1μM 505 Star SNAP ligand (New England Biolabs)) in the medium for more than 30 min. Before observation, medium was changed to the one without ligand and incubated at least 30 min.

### STORM imaging
As the focus depth is limited (−100 nm) because we used STORM imaging in the total reflection microscopy setup to remove signals outside of the focal plane, we had to choose the Golgi in the en face views because we wanted to focus on the localisation of these enzymes within the plane of one cisterna. As more than three dyes cannot be practically used for STORM simultaneously, we used giantin for the first dye and one of the glycosylation enzymes (GALNT6, MGAT1, and XYLT2) for the second dye as described in the text.

Thus, the cell preparation was incubated with the primary antibodies as described above and the secondary antibodies conjugated with CF 568 (Biotium) or Alexa Fluor 647 (Invitrogen). To examine the photoswitching activity of fluorophores, the cell preparation was then mounted with a freshly made STORM buffer: a mixture of 7 mL of GLOX solution (glucose oxidase 70 mg/mL, catalase 3.4 mg/mL in 10 mM Tris, pH 8.0 containing 50 mM NaCl, 70 mL of 1 M cysteamine solution containing 0.74% HCl and 620 mL of 50 mM Tris, pH 8.0 containing 10 mM NaCl, 10% Glucose. Images were captured using a Nikon Eclipse Ti-E inverted microscope (Nikon Instruments Inc.) equipped with an N-STORM superresolution microscope system (Nikon Instruments Inc.) and an ORCA-Flash 4.0 sCMOS camera (Hamamatsu Photonics, Japan). Samples were excited with a 561 nm laser for CF 568 and a 640 nm laser for Alexa 640. The fluorescence emission of CF 568 and Alexa Fluor 647 were collected onto a CFI SR HP Apochromat TIRF 100XC Oil lens (Nikon Instruments Inc.). Images were recorded, and data were acquired using NIS-Elements imaging software (Nikon Instruments Inc.).

### Measuring the areas of zones from STORM image
The detection of individual fluorescent spots in the images for 10,000 frames was performed using the ThunderSTORM plugin in ImageJ[25] with "wavelet filtering" (B-spline order = 2 and B-spline scale = 2.0) and "least squares" (fitting radius = 3 pixels, initial sigma = 1.6 pixels). The x and y coordinates for the detected spots were included in the CSV data output, which was subsequently imported into the SR-Tesseler software[26] for cluster analysis based on Voronoi polygon density methods. To generate Voronoi polygons without correcting for blinking and multi-ON frame detections, the "Detection cleaner" function was used (within the entire ROI). Golgi contours were identified by utilising the "Objects" function with a Voronoi polygon density factor of 2, while cluster contour detection was performed using the "Clusters" function with a density factor of 1. To ensure that size estimations were precise, clusters were filtered according to a

minimum number of localisations of 100. Finally, the output data from SR-Tesseler were plotted and subjected to statistical analysis using ORIGIN 2018b software.

## Expression of the soluble form of XYLT2 and NDST1

The expression plasmids (pEF-BOS/IP empty, pEF-BOS/IP-XYLT2, pEF-BOS/IP-XYLT2-3PA, pEF-BOS/IP-NDST1, and pEF-BOS-NDST1-9xMyc) (2.5 mg) were transfected into COS7 cells on 100-mm dishes using Lipofectamine 3000 according to the instructions provided by the manufacturer. Two days after transfection, 1.5 ml of the culture medium was collected and incubated with 10 ml of IgG-Sepharose for 2 h at 4 °C. The beads recovered by centrifugation were washed three times with 20 mM Tris-HCl (pH 7.5) containing 0.15 M NaCl, 1 mM EDTA, 10% glycerol, and 1% Nonidet 40.

## Immunoblotting of XYLT2 and NDST1 before enzyme assay

The protein A-fused XYLT2, XYLT2-3PA, NDST1, and NDST1-9xMyc bound to IgG-Sepharose beads were mixed with 20 ml of sample buffer solution with reducing reagent for SDS–PAGE and heated at 94 °C for 3 min. Each sample was separated by Bullet PAGE One Precast Gel (5-15%), transferred to AmershamTM HybondTM-PVDF filters, and incubated overnight at 4 °C with rabbit IgG (His-Tag rabbit polyclonal antibody; Cell Signalling Technology) for detection of protein A. The bound antibody was detected with anti-rabbit antibody conjugated to horseradish peroxidase.

## Enzymatic assay with XYLT2 and XYLT2-3PA

The chip column used for the following enzyme assay of XYLT2 was prepared. After a small piece of absorbent cotton was pushed into the tip of a pipet chip, a pipet chip was packed with AG-50 W-X2 (H+ form) resin. The chip column was equilibrated with $H_2O$. The reaction mixture for XYLT2 assays (a total volume of 18.03 mL) contained 10 mL of enzyme preparation, 8.03 mL of 25 mM MES-NaOH (pH 6.5), 5 mM KF, 5 mM $MgCl_2$, 5 mM $MnCl_2$, 8.5 mM UDP-[14 C]Xyl (1 × 105 dpm) (American Radiolabelled Chemicals, Inc., USA), and the synthetic peptide Q-E-E-E-G-S-G-G-G-Q-K (1 nmol). The enzyme reaction was performed at 37 °C for 1 h, and 30 mL of 25 mM MES-NaOH buffer (pH 6.5) was added to the reaction products. The reaction products were applied to the packed chip column. The chip column was washed with 1.5 mL of 0.01 M HCl twice, and the wash-1 and wash-2 fractions were collected. Subsequently, the reaction products were eluted with 1.5 mL of 3 M $NH_4OH$ twice, and Elution-1 and Elution-2 fractions were collected. The radioactivity of the effluent fractions was measured using a 2300TR liquid scintillation counter (PerkinElmer, Waltham, MA). Unreacted UDP-[14 C]Xyl was collected in the wash-1 fraction, while the products synthesised by XYLT2 (Q-E-E-E-G-S([14 C]Xyl)-G-G-G-Q-K) were detected in the elution-1 fraction.

## Enzyme assay of NDST1 and NDST1-9xMyc

The reaction mixture for NDST1 assays (a total volume of 200 mL) contained 25 mL of enzyme preparation, 18.5 mL of 50 mM Tris-HCl (pH 7.5) containing 15 mM $MgCl_2$, 6.64 mM PAPS, and 5 mL of 1 mg/mL K5 polysaccharides (GlycoNovo). The enzyme reaction was performed at 37 °C overnight. The reaction products (5 mL) were digested with a mixture of heparinase from Flavobacterium heparinum (1 mIU) (Seikagaku Corp.) and heparitinase from Flavobacterium heparinum (1 mIU) (Seikagaku Corp.) for 4 h at 37 °C. Reactions were terminated at 94 °C for 5 min. The digests were derivatized with the fluorophore 2-aminobenzamide and then analysed by high-performance liquid chromatography (HPLC) as reported previously[56].

## Analysis of GAGs in Caco2 cells

Upon achieving 80% confluency, the growth media were aspirated, and the cells were washed with phosphate-buffered saline (PBS), then changed to serum-free medium, and further incubated for 12 h. GAGs in Caco2 cells were isolated and purified as described previously[57]. Briefly, cell pellets were homogenised and extracted with acetone three times, and air-dried thoroughly. The dried materials were digested with heat-activated actinase E (10% by weight of dried materials) in 0.1 M borate-sodium, pH 8.0, containing 10 mM CaCl2 at 55 °C for 48 h. The samples were adjusted to 5% trichloroacetic acid and centrifuged. The resultant supernatants were extracted with diethyl ether four times to remove trichloroacetic acid, and then neutralised using 20% $NH_4HCO_3$. The aqueous phase containing 5% sodium acetate was adjusted to 80% ethanol and left overnight at −20 °C. The resultant precipitate was dissolved in $H_2O$, and subjected to gel filtration on a PD-10 column (Cytiva, Marlborough, MA) using $H_2O$ as an eluent. The eluates were collected and evaporated to dryness. GAG samples were digested with chondroitinase (Chase) ABC, Heparitinase (HSase), and Heparinase (Hepase) at 37 °C for 4 h. Chase ABC from Proteus vulgaris (EC numbers: 4.2.2.20 & 4.2.2.21), HSase from Flavobacterium heparinum (EC number: 4.2.2.8), and Hepase from Flavobacterium heparinum (EC number: 4.2.2.7) were purchased from Seikagaku Biobusiness Corporation (Tokyo, Japan). The resultant digests were labelled with 2-aminobenzamide, as described previously[56,58], and analysed by high performance liquid chromatography (HPLC) on an amine-bound silica PA-G column (4.6 × 250 mm; YMC Co., Kyoto, Japan) using a linear gradient of NaH2PO4 at a flow rate of 1 ml/min at room temperature as reported previously[56]. Eluates were monitored using an RF-10AXL fluorometric detector (Shimadzu Co., Kyoto, Japan) with excitation and emission wavelengths of 330 and 420 nm, respectively.

## Reporting summary

Further information on research design is available in the Nature Portfolio Reporting Summary linked to this article.

## Data availability

The authors declare that all data supporting the findings of this study are available within the article and its supplementary information or are available from the corresponding author on request. Source data are provided with this paper.

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

## Acknowledgements

This work is dedicated to K.K. who passed away on Apr. 6, 2024. This work was supported by Grants-in-Aid for Scientific Research from the Ministry of Education, Culture, Sports, and Technology of Japan (grant numbers: 17H0622 and 21H02658 to A.H., 21K06734 to M.K. 17H06413 and 17H06420 to K.K. and A.N. 18H05275 to K.K., T.T., and A.N.). We thank K. Toida (Kawasaki Medical University), I. Nishimura (Osaka University), T. Uemura, and S.Waguri (Fukushima Medical University) for advice and assistance with immunoelectron microscopy, A.Satoh (Okayama University), N. Nakamura (Kyoto Sangyo University), F.Perez (Institut Curie, France), and J. Takagi (Osaka University) for sharing materials, S.Hattori (Osaka University) for advice on statistics, M. Zhiyi (Osaka University) and K. Ishii (RIKEN) for cell culture, and C.G. Burd (Yale University) for critically reading the manuscript, and R. Harada drawing the graphic abstract. This study was supported by IEXAS, CentMeRE, and CoMIT Omics Center, Graduate School of Medicine, Osaka University.

## Author contributions

A.H. conceived the project. A.H., M.K., K.K., T.S., and S.Kanda designed and performed experiments. H.K., S.N., and M.N. performed glycosylation analysis. E.M. advised glycosylation analysis. K.G.N.S., K.M.H., S.Kume, and K. T. performed the STORM analysis. K.M., S.Y., M.Y-H., S.G., and Y.Z. constructed plasmids. T.K. generated antibodies. A.H., M.T., and M.H-N. performed electron microscopy. A.H. draughted and edited the manuscript. T.T. and A.N. supervised SCLIM analysis.

## Competing interests

The authors declare no competing interests.
