## [Peer Review File · Nature Communications]

Dynamic movement of the Golgi unit and its glycosylation enzyme zonesREVIEWER COMMENTS

Reviewer #1 (Remarks to the Author):

The organization of the Golgi stack is of fundamental importance for both glycosylation and membrane traffic, and yet remains poorly understood. This technically accomplished paper reports an examination of the distribution of a range of Golgi glycosylation enzymes in mammalian tissue culture cells using both fixed and live cell imaging. They also examine in live cells the distribution of over-expressed fluorescent-protein-tagged forms of two large coiled-coil proteins that surround the rim of the Golgi stack (giantin and golgin84).

The authors make two striking observations. Firstly, they look in Golgi “mini stacks” formed by depolymerizing microtubules and find that the Golgi enzymes are not distributed uniformly but are found clustered in what they term “zones”. Secondly, they find that the normal Golgi ribbons that are present in cells with intact microtubules are actually comprised of closely apposed Golgi units which are surrounded by giantin and golgin84. The latter two proteins are proposed to hold the Golgi units together and to allow transport between them to ensure efficient distribution of enzymes or cargo between the Golgi units. If this is disrupted, then glycosylation is defective.

The work is of a high technical quality with care taken to tag the glycosylation enzymes in the genome to avoid issues of over-expression (although this is not done with giantin and golgin-84 as discussed below). In addition, the analysis is extensive and carefully quantified. The formation of zones of enzymes in living cells is interesting as this has only been previously observed in fixed cells. Likewise, the movies of cells expressing fluorescent-protein-tagged giantin or golgin-84 are particularly striking as they suggest that the Golgi “ribbon” is actually a conglomeration of individual Golgi units. Thus, the paper is likely to be of a lot of interest to the Golgi field.

However, one issue is that some of the key conceptual conclusions in this paper have been reported previously in a study from Adam Lindstedt’s lab:

Puthenveedu et al (2006) GM130 and GRASP65-dependent lateral cisternal fusion allows uniform Golgi-enzyme distribution. *Nature Cell Biology* 8:238-248.

This work is not cited and so the authors may have been unaware of it. In this previous study the authors also break up the Golgi ribbon by removing a golgin (in their case GM130) and also use photobleaching to show that having the Golgi dispersed into ministacks reduces diffusion of enzymes between individual units and this causes defects in glycosylation.

This new paper contains is much more extensive in its analysis, and examines the distribution of Golgi enzymes into zones, and so extends the previous work and adds further rigour. But given that the conclusions are not quite a novel as the authors suggest, then I feel that it is particularly important to make sure that the current study is as technically sound as possible, and the findings are properly discussed in the context of what has been previously reported. Below I have listed some suggestions for

this, and if these were addressed I would be supportive of publication.

1) Obviously, the authors need to cite and discuss the Puthenveedu et al paper mentioned above.

2) The authors should cite and discuss the paper from Fourriere et al (PubMed ID 27411366), that reports that many of the Golgi mini-stacks formed in nocodazole treated HeLa cells are not fully functional for secretion. Have they determined whether the mini-stacks or Golgi units they are examining are all equally functional?

3) The authors conclude that the Golgi ribbon consists of Golgi units with similar dimensions to those seen in nocodazole treated cells (Figure 3). However, the images that they show are not entirely consistent with this. In some places the Golgi has areas where the giantin staining marks out a long oval where the long dimension is several times the short dimension (eg HeLa and MEF images in Figure 3c). This is surely very different to the Golgi units seen in nocodazole-treated cells which are generally circular.

4) Related to point (3), it is possible that the striking movies of linked Golgi units in the Golgi ribbon are an artefact of the over-expression of giantin that is used to visualize them (Figure 4 and associated movies). It is possible that the excess giantin spreads around the rims and causes larger ribbons to fragment. To address this the authors should do the same as they did for the enzymes and attached the nNeonGreen to the endogenous giantin by integration. Such tagging should be possible as it is used in the Fourriere et al paper mentioned above. This approach would then provide unambiguous evidence that the Golgi organization that they see exists in unperturbed cells.

5) The authors should discuss if they have evidence if the tagged forms of giantin and golgin84 are functional.

6) The authors provide very interesting live cell imaging of Golgi enzymes using tagging of the endogenous genes. The movements are very rapid, but the frames shown are taken three seconds apart. Would it be possible to also image at a higher frame rate, perhaps in a smaller area for a shorter time, to reveal these rapid movements more clearly.

7). Some of the enzymes are located around the Golgi rim, and can appear punctate. The authors should also stain the Golgi stack for COPI to see if these puncta correspond to COPI-coated buds or vesicles. In this context, it is rather surprising that the paper contains no mention of COPI or coatamer, when Golgi resident enzymes are known to be recycling in these vesicles and so some of the population of enzymes will always be in these buds and vesicles and not just in the cisternae. This must be discussed properly.

8) The movies are very impressive. It would greatly help viewers if each movie could have a starting frame that explains what is shown along with the frame rate.

Reviewer #2 (Remarks to the Author):

In this study, Harada et al. investigate the basic structure and dynamics of the Golgi apparatus and its glycosylation enzymes with unprecedented high spatiotemporal resolution to better understand the essential mechanism of Golgi glycosylation. Previous high-resolution studies of the lateral distribution of glycosylation enzymes using Airyscan imaging have been methodologically limited by the use of image averaging, which is not suitable for accurate protein localization and area measurement, and by the overexpression of tagged glycosylation enzymes, which has been shown to alter their normal cellular distribution. To overcome these limitations, the authors endogenously expressed tagged glycosylation enzymes using the CRISPR knock-in method and employed higher spatial resolution light microscopy systems such as STORM or SCLIM. More importantly, they took advantage of the high-speed capacity of SCLIM microscopy to address the dynamics of the structural Golgi units (stacks of Golgi cisternae) and glycosylation enzymes with super-resolution live imaging. The authors find that the Golgi ribbon is assembled by dynamic units, which are surrounded by giantin and dynamically change shape through fission and fusion events. They also find that early-stage glycosylation enzymes form mobile small domains or zones near the rim of a single cis or medial cisterna. By contrast, zones of later-stage glycosylation enzymes occupy a more central position within the trans cisterna. Interestingly, photobleaching analysis suggests that mobile zones of early-stage N- and O-glycosylation enzymes are colocalized and remain within the same cisterna, whereas mobile zones of early-stage glycosaminoglycan (GAG) synthesizing enzymes can travel between different Golgi units through the connecting tubules, possibly due to their smaller size. Finally, the authors show that the depletion of giantin dissociates the Golgi units and interferes with the movement of GAG-synthesizing enzymes between them, leading to a defect in GAG synthesis.

Overall, this is a high-quality and technically excellent study that takes advantage of powerful super-resolution live imaging to address for the first time at high spatiotemporal resolution the dynamics of Golgi units and their endogenously expressed glycosylation enzymes, providing the structural basis of essential Golgi function. The study also identifies giantin as a factor involved in the clustering of Golgi units and required for correct localization and function of early-stage GAG-synthesizing enzymes, which may help to better understand the link between the functional structure of the Golgi and human pathogenesis, although the underlying mechanism remains to be elucidated. The findings and conclusions of this study are relevant and fully supported by the results.

Specific comments:

1. FRAP analysis suggests that the early-stage GAG-synthesizing enzyme XYLT2 can move rapidly between Golgi units when they are clustered, but not when they are dissociated. This result may explain why giantin knockdown cells show an aberrant distribution of XYLT2 in the dissociated Golgi units and consequently a reduced synthesis of mature GAGs. To support this possibility, the authors could determine whether nocodazole-treated cells also show a defect in mature GAG synthesis and an aberrant distribution of XYLT2 in their dissociated Golgi units.
2. Figure 6e shows that the amount of mature GAGs was reduced in giantin knockdown cells. However, since the methodology used to obtain this result is not explained, it is not clear whether the immature GAGs produced are secreted or trapped within the dissociated Golgi units. This should be clarified for proper interpretation of the data.

3. The use of SCLIM microscopy in this study allowed the observation of the dynamic distribution of Golgi glycosylation enzymes at nanoscale resolution. Interestingly, the authors find that early glycosylation enzymes form highly mobile small domains or zones that move rapidly around the rim of a single cis or medial cisterna, whereas zones of late glycosylation enzymes occupy a more central position within a trans cisterna. I would like to ask the authors to briefly discuss the possible mechanistic implications of these relevant findings.

4. Extended Figure 4e. "Two-colour SCLIM imaging..." should be "three-colour SCLIM imaging..."

Reviewer #3 (Remarks to the Author):

In this Manuscript Harada and colleagues use a combination of gene editing and advanced microscopy techniques to study the localization of Golgi glycosylation enzymes. They generate KI cell lines of various enzymes with various tags to be able to perform live-cell microscopy and look at their localization within the Golgi stacks. They perform STORM, airyscan and super-resolution confocal live imaging microscopy (SCLIM). They suggest that the various enzymes occupy segregated nano-domains or "zones" and that the Golgi is composed of "units" that are connected laterally to form a Golgi ribbon.

While the approach is novel and promises to reveal great insights into Golgi organization and functioning, I find some of the conclusions not very well supported by the data. The conclusions about the organization of the Golgi in units and with giantin having a functional role is not novel and weak and only supported by 2D live-cell microscopy (Figure 4a-b). I suggest the authors focus on the novel and exciting part of the paper (Fig 1 and 2) which deals with the super-resolved localization of Golgi enzymes. If the authors want to make a claim of the exchanges of Golgi units and a role of Giantin in linking the cisternae via tubules more imaging data would be necessary (multi-color 3D with multiple markers to capture the twisting and 3D conformation of the Golgi ribbon. Time lapses after addition and removal of nocodazole to see disassembly and assembly.

In conclusion I find the work exciting and I am happy to re-review the manuscript after major revisions and re-writing.

Apart from what stated above, here are my comments point by point:

- In general a better introduction of which imaging technique and the rationale for specifically using that is missing. I am not familiar with SCLIM. I do not know what resolution can be achieved in all directions and that is very important to interpret the Golgi imaging data.

Example: Figure 4 in a-b Airy scan is used while in c SCLIM is used... one is 2D and the other 3D... In general imaging modalities and acquisition parameters should be better described in the figure legends.

- Hard to follow the logics and the text. Why in figure 2 the authors first use STORM for looking at early enzymes and then SCLIM for later enzymes?

- Extended data Fig 1. I do not understand the strategy used for KIs. Is the cassette for Hygro resistance immediately downstream of the tag and stop codon? Does the disruption of the endogenous polyA affect protein levels? It is hard to say without a proper loading control for the western blots...

- Positive controls for all the imaging techniques are missing. For example the same Halo KI labelled with

2 dyes should be used. For the STORM experiments the same primary antibody should be labelled with different secondaries. This is important to interpret the co localization data

- Graphs are often missing details about axis etc... Extended data 4b = % of what? 4g top and bottom graphs would be sep explanatory with a better explanation of the axis. For example: minutes from addition of Biotin. Not everybody knows how RUSH works...
- Line 188. The authors talk about 1-3 microns size units. I find it hard to understand what a unit is and how the graph in Figure 3b was calculated... Also in line 193, they claim the units are connected by tubules. This is also not very clear to me. I suggest the authors remove this conclusion as this is not supported.
- Figure 4. The authors claim this units undergo fission and fusion. As this is 2D microscopy this is very hard to conclude and an overinterpretation in my opinion as only one marker is visualized
- Figure 5. Again it is very hard to understand the interpretation of the authors who claim that the enzyme is moving along the cisternae. Is this 3D imaging? Framerate?
Graphs in C are poor quality and small and lack explanatory titles and axes

- Figure 6. I believe that the notion that giantin may help link Golgi ministacks and the KD triggers fragmentation into ministacks as nocodazole does it not novel.

<https://pubmed.ncbi.nlm.nih.gov/31544102/>

<https://pubmed.ncbi.nlm.nih.gov/23555793/>

Some groups have shown that Giantin KO does not have any effect on Golgi morphology

<https://www.ncbi.nlm.nih.gov/pmc/articles/PMC5769581/>

Minor points

- What's a PA tag? Line 77 describe abbreviation
- Extended data fig 3c. Better labelling to show that Giantin is not the KI but an antibody! Also as giantin is big it would be useful to know where the epitope binds etc... considering the authors use this as a cisternal rim marker from now on
- Figure 1. The author say in panel f "3D co-localization analysis". Does it mean that all the imaging was acquired in 3D? This should be started in the legends for all figures.
- I have to admit that all the figures looked like some kind of processing and manipulation was applied. Which makes it hard to interpret the data. All the changes made to the raw images should be, in my opinion, openly stated in the legends
- Figure 2a. The authors should state that they picked face ministacks?
- Figure 3. Are cells alive or dead? This should also be stated in the legends

Reviewer #1 (Remarks to the Author):

1) Obviously, the authors need to cite and discuss the Puthenveedu et al paper mentioned above.

Thank you for the kind and valuable instruction.

We cited this paper as reference 43 and added discussion related to this subject in lines 384-386.

2) The authors should cite and discuss the paper from Fourriere et al (PubMed ID 27411366), that reports that many of the Golgi mini-stacks formed in nocodazole treated HeLa cells are not fully functional for secretion. Have they determined whether the mini-stacks or Golgi units they are examining are all equally functional?

Thank you for the kind and valuable instruction.

Their paper reports that many of the Golgi mini-stacks formed in nocodazole treated HeLa cells are not fully functional for secretion. To distinguish the effect of Golgi dispersion from microtubule depolymerization, they expressed a dominant-negative dynein construct, p150-CC1, to disperse the Golgi without microtubule depolymerization. They observed a normal transport of TNF1 in Golgi dispersed cells by p150-CC1 overexpression. However, when they further treated nocodazole to p150-CC1 expressed cells, they observed blockade of TNF1 transport.

Our giantin knockdown or knockout cells seems similar to p150-CC1 expressing cells because microtubules are not depolymerized. Thus, ministacks caused by giantin depletion are likely to be competent for cargo transport to the plasma membrane. On the other hand, we observe greater divergence in the amount of XYLT2 among the isolated Golgi units in the absence of giantin (Fig. 8d). This indicates that cargos for the plasma membrane can go through the Golgi units generated by giantin depletion although some of them acquire insufficient glycosylation of GAG because of insufficient amount of its enzymes. Therefore, in that sense, units which have little amount of XYLT2 are not fully functional in terms of GAG synthesis.

We have added the discussion above in lines 350-362.

3) The authors conclude that the Golgi ribbon consists of Golgi units with similar dimensions to those seen in nocodazole treated cells (Figure 3). However, the images that they show are not entirely consistent with this. In some places the Golgi has areas where the giantin staining marks out a long oval where the long dimension is several times the short dimension (eg HeLa and MEF images in Figure 3c). This is surely very different to the Golgi units seen in nocodazole-treated cells which are generally circular.

Thank you for the valuable comment.

As this reviewer pointed out, there are elongated Golgi units in fixed cells. We changed our description as such and added the description below in lines 212-215.

'Sometimes we observed elongated units in some type of cells. Considering the dynamic nature of the units, this may due to the elongation of the units caused by tension between them and/or reflect the transient shape after deformation and fusion as shown in Figure 5 and supplementary movies 4-8.'

4) Related to point (3), it is possible that the striking movies of linked Golgi units in the Golgi ribbon are an artefact of the over-expression of giantin that is used to visualize them (Figure 4 and associated movies). It is possible that the excess giantin spreads around the rims and causes larger ribbons to fragment. To address this the authors should do the same as they did for the enzymes and attached the nNeonGreen to the endogenous giantin by integration. Such tagging should be possible as it is used in the Fourriere et al paper mentioned above. This approach would then provide unambiguous evidence that the Golgi organization that they see exists in unperturbed cells.

Thank you for raising an important point.

We have generated EGFP-Giantin knockin Caco2 cells using the same knockin construct in the paper by Fourriere et al that was provided from the corresponding author of this paper. As we detected the signal by immunofluorescence against EGFP, knockin was successful, but endogenous EGFP expression was apparently insufficient for live imaging. In parallel, as we knew visualizing EGFP for live imaging was hard in our experience, we made the knockin construct by ourselves.

As tagging at the N terminus is technically hard to select knockin clones within this limited time period, we inserted Halo tag at the C terminus. Ideally, it is better to make giantin knockin cells. However, we were not able to make giantin-Halo cDNA which is necessary to check the localization of tagged construct. On the other hand, golgin84-Halo cDNA was successfully generated and colocalized with endogenous Golgi rim markers. Afterwards, we succeeded in making Golgin84-Halo knockin clones and performed live imaging. We added the result in Supplementary Fig. 6 and Supplementary movies 7 and 8 (lines 237-239). Here, we observed essentially the same behavior of Golgi units, such as fusion, separation, attachment, and detachment as in figures (Fig. 5, 6) and movies (Supplementary movies 4-7) which were obtained by overexpression of giantin and golgin84.

5) The authors should discuss if they have evidence if the tagged forms of giantin and golgin84 are functional.

Thank you for the valuable comment.

A previous paper showed the phenotypes of giantin and golgin84 knockdown or knockout.

The paper by Stevenson (ref.42) showed a decrease in GALNT3 expression in giantin knockout RPE-1 cells. However, when we knocked down giantin (to avoid clonal variation) in RPE-1, we observed an increase in the amount of GALNT3 mRNA. Therefore, we considered this phenotype might not be reliable as a consequence of giantin depletion.

For golgin84, Sohda et al. showed a decrease in glycosylation and a decrease in the molecular weight of lamp1 by golgin84 siRNA treatment of HeLa cells (*Traffic* 11: 1552–1566, 2010). However, we did not observe a decrease in the molecular weight of lamp1 in golgin84 knockdown Caco2 and HeLa cells in our hands. For these reasons, we were not able to do rescue experiments using these standards with our giantin and golgin84 constructs.

However, at least, as mNeonGreen-giantin construct can rescue the Golgi fragmentation phenotype in Fig.8a, b, this construct seems functional morphologically. Also, the behaviour of the Golgi units, such as fusion and separation, was similar between mNeonGreen-giantin transfected cells and mNeonGreen-golgin84 transfected cells (Supplementary Table 1).

In addition, the shape of the Golgi in golgin84-Halo knockin Caco2 cells appeared similar to the parental Caco2 cells (for example, please compare Figure 4a and Supplementary Fig. 6c). From these observations, at least, it is safe to say that the behaviour of the Golgi using these constructs largely reflect their endogenous behaviour.

We added this description in lines 305-312.

6) The authors provide very interesting live cell imaging of Golgi enzymes using tagging of the endogenous genes. The movements are very rapid, but the frames shown are taken three seconds apart. Would it be possible to also image at a higher frame rate, perhaps in a smaller area for a shorter time, to reveal these rapid movements more clearly.

Thank you for raising the important point.

We were also curious to see the dynamics of units in higher time resolution. However, it is technically difficult to increase the frame rate by reducing the range of Z axis or the scanning area in XY plane. We also tried to increase the frame rate by fixing the focus to take the image at video rate (60 frames/sec). Though it seems fast enough to capture the movement, we were not able to observe the movement of small areas or zones which are likely to be visualized only by 3D reconstruction. Therefore, it seems almost impossible using the current technique and it should be solved in future by developing faster SCLIM. This description was added in lines 249-252.

7). Some of the enzymes are located around the Golgi rim, and can appear punctate. The authors should also stain the Golgi stack for COPI to see if these puncta correspond to COPI-coated buds or vesicles. In this context, it is rather surprising that the paper contains no

mention of COPI or coatomer, when Golgi resident enzymes are known to be recycling in these vesicles and so some of the population of enzymes will always be in these buds and vesicles and not just in the cisternae. This must be discussed properly.

Thank you for the important comment.

In response to this, we stained and compared the localization of glycosylation enzymes, giantin, and β COP by SCLIM. Localisation of GALNT6 was also shown by immunoEM. These results are shown in Fig. 3 and described in lines 190-199.

8) The movies are very impressive. It would greatly help viewers if each movie could have a starting frame that explains what is shown along with the frame rate.

Thank you for the comment.

We added the starting or inserted frames with necessary information in all supplementary movies.

Reviewer #2 (Remarks to the Author):

Specific comments:

1. FRAP analysis suggests that the early-stage GAG-synthesizing enzyme XYLT2 can move rapidly between Golgi units when they are clustered, but not when they are dissociated. This result may explain why giantin knockdown cells show an aberrant distribution of XYLT2 in the dissociated Golgi units and consequently a reduced synthesis of mature GAGs. To support this possibility, the authors could determine whether nocodazole-treated cells also show a defect in mature GAG synthesis and an aberrant distribution of XYLT2 in their dissociated Golgi units.

Thank you for the valuable comment.

According to this comment, we dissociated the unit by nocodazole. We successfully observed both the increased divergence in the amount of XYLT2 per area of unit and a reduction in the amount of mature GAG by nocodazole treatment. We added these results in Supplementary Fig.9 and described in lines 289-293.

We also modified the graph to compare the distribution of the signals more easily by showing the value of variance in signal distribution in each cell instead of collecting all values from all cells by violin plot (Fig. 8d and Supplementary Fig. 9b). Though we changed the mode of display, the result of statistics was the same as the previous one. The previous and the current

data are both based on the same raw data in Supplementary Fig. 10.

2. Figure 6e shows that the amount of mature GAGs was reduced in giantin knockdown cells. However, since the methodology used to obtain this result is not explained, it is not clear whether the immature GAGs produced are secreted or trapped within the dissociated Golgi units. This should be clarified for proper interpretation of the data.

We deeply apologize for the insufficient explanation.

They are heparan sulfate GAGs trapped in knockdown cells. According to this comment, we added technical explanations in the methods section (lines 765-786) and in the figure legends of Fig.8e (lines 1167-1168).

3. The use of SCLIM microscopy in this study allowed the observation of the dynamic distribution of Golgi glycosylation enzymes at nanoscale resolution. Interestingly, the authors find that early glycosylation enzymes form highly mobile small domains or zones that move rapidly around the rim of a single cis or medial cisterna, whereas zones of late glycosylation enzymes occupy a more central position within a trans cisterna. I would like to ask the authors to briefly discuss the possible mechanistic implications of these relevant findings.

Thank you for the valuable comment.

According to this comment, we proposed one possible explanation in lines 337-344.

4. Extended Figure 4e. "Two-colour SCLIM imaging..." should be "three-colour SCLIM imaging..."

Thank you for the comment. We corrected this part (line 1270).

Reviewer #3 (Remarks to the Author):

I numbered the comments to facilitate discussion.

Comment 1

The conclusions about the organization of the Golgi in units and with giantin having a functional role is not novel and weak and only supported by 2D live-cell microscopy (Figure 4a-b). I suggest the authors focus on the novel and exciting part of the paper (Fig 1 and 2) which deals with the super-resolved localization of Golgi enzymes. If the authors want to make a claim of the exchanges of Golgi units and a role of Giantin in linking the cisternae via tubules more imaging data would be necessary (multi-color 3D with multiple markers to capture the twisting and 3D conformation of the Golgi ribbon).

Thank you for the valuable comment.

As we have taken pictures in three dimensions by SCLIM, we were able to show twisting and 3D conformation of the Golgi ribbon by changing the angle in Supplementary movie 1, 3, 8, 10 and supplementary Fig. 5. Also, we added that the term '3D' or 'three dimensions' in the text and figure legends as much as possible (e.g., lines 203, 1014, 1025, etc.) as well as '2D' in case we use Airyscan (e.g., lines 1094, 1131, 1143, etc.).

Comment 2

Time lapses after addition and removal of nocodazole to see disassembly and assembly.

According to this advice, we made 3D time-lapse observations by SCLIM to show disassembly and assembly of the Golgi (Supplementary movie 2, 3) (lines 218-227). As shown in these movies, after nocodazole removal, we were able to observe reconstruction of the Golgi ribbon by units through tubules between them.

Apart from what stated above, here are my comments point by point:

Comment 3

• In general a better introduction of which imaging technique and the rationale for specifically using that is missing. I am not familiar with SCLIM. I do not know what resolution can be achieved in all directions and that is very important to interpret the Golgi imaging data. Example: Figure 4 in a-b Airy scan is used while in c SCLIM is used... one is 2D and the other 3D... In general imaging modalities and acquisition parameters should be better described in the figure legends.

We apologize for insufficient explanation of SCLIM.

We added information of microscopies used and which is 2D and 3D as described above. We added information of spatial resolution and other parameters of SCLIM in lines 670-681 according to our previous paper in this journal (ref 8) as well as in the figure legends (lines 1030-1031, 1058-1059, etc.).

Comment 4

- Hard to follow the logics and the text. Why in figure 2 the authors first use STORM for looking at early enzymes and then SCLIM for later enzymes?

Thank you for the valuable comment.

We mainly used SCLIM to identify the localization of zones of each enzyme. We used STORM to quantify differences in the area of each zone. In case of late-stage enzymes, N-glycosylation and O-glycosylation are mediated by the same enzyme, β 4GalT1. Therefore, we did not use STORM for the late-stage enzyme in this paper. We added this description in lines 167-168.

Comment 5

- Extended data Fig 1. I do not understand the strategy used for KIs. Is the cassette for Hygro resistance immediately downstream of the tag and stop codon? Does the disruption of the endogenous polyA affect protein levels? It is hard to say without a proper loading control for the western blots...

Thank you for the valuable comments.

We changed the illustration to better illustrate the knockin construct (Supplementary Fig. 1b). We also added loading control in Supplementary Fig. 1d.

Comment 6

- Positive controls for all the imaging techniques are missing. For example the same Halo KI labelled with 2 dyes should be used. For the STORM experiments the same primary antibody should be labelled with different secondaries. This is important to interpret the colocalization data

Thank you for the valuable comments.

We added positive controls as indicated by this reviewer (Supplementary Fig.3b for SCLIM and Fig. 2b for STORM).

Comment 7

• Graphs are often missing details about axis etc... Extended data 4b = % of what? 4g top and bottom graphs would be self-explanatory with a better explanation of the axis. For example: minutes from addition of Biotin. Not everybody knows how RUSH works...

We apologize for insufficient description.

We added description in the Y axis of Supplementary Fig. 4b and in the X axis of Supplementary Fig. 4g.

Comment 8

• Line 188. The authors talk about 1-3 microns size units. I found it hard to understand what a unit is and how the graph in Figure 3b was calculated...

We just measured the longer diameter in the circular to elongated structure encircled by giantin (or TMF1 when giantin is knocked down) as shown in the picture below (left: before measurement, right: after measurement). Length of each yellow line in the right panel was measured and its distribution was shown in Fig. 4b (former Fig. 3b). We added the following description as a definition of a unit in the figure legend of Fig. 4b (lines 1087-1088). 'One unit is defined as circular to elongated area encircled by giantin within which contains glycosylation enzymes.'

Comment 9

Also in line 193, they claim the units are connected by tubules. This is also not very clear to

me. I suggest the authors remove this conclusion as this is not supported.

Thank you for the valuable comment.

Concerning the tubules between units, it seems clear that there are tubules between units in Supplementary movie 1 and 8 by showing the Golgi from various angles. Tubules between the Golgi discs were presented in the previous literature using EM tomography (ref. 34). This finding also supports our observation by SCLIM.

Comment 10

- Figure 4. The authors claim this units undergo fission and fusion. As this is 2D microscopy this is very hard to conclude and an overinterpretation in my opinion as only one marker is visualized

Thank you for the valuable comment.

To more clearly show the fusion, separation, attachment, and detachment, we used SCLIM for observation and confirmed these processes by observation from various angles in 3D (Supplementary movie 5, 8, 9).

Comment 11

- Figure 5. Again it is very hard to understand the interpretation of the authors who claim that the enzyme is moving along the cisternae. Is this 3D imaging? Framerate?

Thank you for the valuable comment.

To show the movement of the glycosylation enzymes more clearly, we used SCLIM and confirmed these processes by observation from various angles in 3D (Supplementary movie 9, 10, 12). Frame rates (speed) are shown in the starting frame in each movie.

Comment 12

Graphs in C are poor quality and small and lack explanatory titles and axes

Thank you for the comment. We enlarged the graph and added descriptions (Fig. 7c) (lines 1142-1147).

Comment 13

- Figure 6. I believe that the notion that giantin may help link Golgi ministacks and the KD triggers fragmentation into ministacks as nocodazole does it not novel.

<https://pubmed.ncbi.nlm.nih.gov/31544102/>

<https://pubmed.ncbi.nlm.nih.gov/23555793/>

Thank you for the valuable information.

There are a number of golgins (e.g., GCC2: Traffic 8:758; golgin84: JCB 160:201,2003; GM130: ref42; GRASPs: refs43-45; GMAP210: JCB 145:83, 1999; GCC1: JCS: jcs211987, 2018; TMF1: BMC Cell Biol 5:18, 2004) known to fragment the Golgi by knockdown, knockout, or overexpression.

Though our finding may not be a new one, we are happy that our finding on the importance of giantin than other golgins in linking the unit is supported by these papers. In addition, we observed the similar fragmentation phenotype in HeLa cells and Caco2 cells, suggesting the universal significance of giantin for the architecture of the Golgi.

Comment 14

Some groups have shown that Giantin KO does not have any effect on Golgi morphology
<https://www.ncbi.nlm.nih.gov/pmc/articles/PMC5769581/>

Thank you for the valuable information. Their results might reflect the size of the unit rather than the size of the Golgi ribbon because the diameter of the Golgi in this article distributed from 0.5 to 2 μ m which corresponds to the distribution of the Golgi unit in our paper (Fig.4b) which is relatively unchanged regardless of the presence of giantin.

Minor points

Comment 15

- What's a PA tag? Line 77 describe abbreviation.

Thank you for the comment. We added information on this tag in the text (lines 80-82).

Comment 16

- Extended data fig 3c. Better labelling to show that Giantin is not the KI but an antibody!

We apologize for making confusion. We modified the figure to improve labeling (Supplementary Fig. 3c).

Comment 17

Also as giantin is big it would be useful to know where the epitope binds etc... considering the authors use this as a cisternal rim marker from now on

Thank you for the valuable comment. We added information on the epitopes in the method section (lines 593-594).

Comment 18

- Figure 1. The author say in panel f "3D co-localization analysis". Does it mean that all the imaging was acquired in 3D? This should be started in the legends for all figures.

We apologize for insufficient description. We added '3D' in the figure legends when we took pictures in 3D (answered in comment #1).

Comment 19

- I have to admit that all the figures looked like some kind of processing and manipulation was applied. Which makes it hard to interpret the data. All the changes made to the raw images should be, in my opinion, openly stated in the legends

We apologize for insufficient explanation of SCLIM. We added information of microscopies used. We added information of spatial resolution and other parameters of SCLIM in lines 670-681 according to our previous paper in this journal (ref 8) (answered in comment #3).

Comment 20

- Figure 2a. The authors should state that they picked face ministacks?

Thank you for the comment. We added the description that we chose ministacks in the text (line 1034).

Comment 21

- Figure 3. Are cells alive or dead? This should also be stated in the legends

Thank you for the comment. We added the description that the cells are fixed in the text (line 1079).

REVIEWERS' COMMENTS

Reviewer #1 (Remarks to the Author):

The authors have done an excellent job of engaging with my comments and suggestions. I also feel that they have done the same for the other reviewers. This work is a significant advance in our understanding of the Golgi, and so it was important that the data is as robust as possible, and the authors have now achieved this.

I have two suggestions:

a) The authors have added helpful title frames to all the movies. For the two colour movies they explain what each colour represents, but for a couple of the single colour movies this is not stated and it would be helpful for viewers to know what marker is being shown.

b) Given the significance of this study, would it be possible for the authors to be offered the option of it being 'promoted' to Nature Cell Biology?

Reviewer #2 (Remarks to the Author):

The authors have adequately addressed all my comments by performing the proposed experiment, adding methodological information, and discussing possible mechanistic implications of their findings. I have no further comments for them. I believe that this is a high quality and excellent study that makes a relevant contribution to the field.

Reviewer #3 (Remarks to the Author):

I am happy with the authors revision. Methods are now much clearer and it is easier to follow what was done and how.

I have just one small clarification but I do not need to see the manuscript again. In the figure legends the authors are now stating frame rates for 3D SCLIM imaging. Example in Figure 6 they say "frame rate is 3 seconds/frame". I believe this is a whole volume composed of 31-36 optical sections? So 3 seconds/volume or stack?

To reviewer #1

>The authors have added helpful title frames to all the movies. For the two colour movies they explain what each colour represents, but for a couple of the single colour movies this is not stated and it would be helpful for viewers to know what marker is being shown.

According to this advice, we added title frames to single colour movies (Supplementary movies 2, 3, 4, and 6).

To Reviewer #3

> I have just one small clarification but I do not need to see the manuscript again. In the figure legends the authors are now stating frame rates for 3D SCLIM imaging. Example in Figure 6 they say "frame rate is 3 seconds/frame". I believe this is a whole volume composed of 31-36 optical sections? So 3 seconds/volume or stack?

According to this advice, we added the description (3 seconds/volume) in lane 1108 and 1118 and in Supplementary figure 5c (10 seconds/volume).